# FORWARD LEARNING WITH DIFFERENTIAL PRIVACY

## ABSTRACT

Differential privacy (DP) in deep learning is a critical concern as it ensures the confidentiality of training data while maintaining model utility. Existing DP training algorithms provide privacy guarantees by clipping and then injecting external noise into sample gradients computed by the backpropagation algorithm. Different from backpropagation, forward-learning algorithms based on perturbation inherently add noise during the forward pass and utilize randomness to estimate the gradients. Although these algorithms are non-privatized, the introduction of noise during the forward pass indirectly provides internal randomness protection to the model parameters and their gradients, suggesting the potential for naturally providing differential privacy. In this paper, we propose a privatized forward-learning algorithm, Differential Private Unified Likelihood Ratio (DP-ULR), and demonstrate its differential privacy guarantees. DP-ULR features a novel batch sampling operation with rejection, of which we provide theoretical analysis in conjunction with classic differential privacy mechanisms. DP-ULR is also underpinned by a theoretically guided privacy controller that dynamically adjusts noise levels to manage privacy costs in each training step. Our experiments indicate that DP-ULR achieves competitive performance compared to traditional differential privacy training algorithms based on backpropagation, maintaining nearly the same privacy loss limits.

## 1 INTRODUCTION

Deep neural networks have made substantial advancements across various domains, such as image recognition (Meng et al., 2022; Dosovitskiy et al., 2020; Zhao et al., 2020), natural language processing (Deng & Liu, 2018; Meng et al., 2022; Han et al., 2021), and autonomous driving (Huang et al., 2018; Häuslschmid et al., 2017; Chen et al., 2015). However, training these powerful models often involves vast amounts of data, including personal data gathered from the Internet, exacerbating privacy concerns. It has been well-documented that neural networks do not merely learn from data but can also memorize specific instances (Carlini et al., 2019; 2021).

Differential privacy (DP) has emerged as a widely accepted metric for assessing the leakage of sensitive information in data (Liu et al., 2024a). In the realm of model training, DP mechanisms (algorithms) aim to ensure that the presence or absence of any single data sample does not significantly influence the learned parameters. The most popular learning algorithm, Differentially Private Stochastic Gradient Descent (DP-SGD) (Abadi et al., 2016), employs a typical strategy to safeguard privacy: assessing algorithms' sensitivity and introducing randomness by adding random noise to their final output. The manually introduced randomness breaks the deterministic of the computed gradient and protects privacy. However, there are many problems when deploying DP-SGD. First, it needs to compute the gradient of each sample individually, causing huge time consumption compared to traditional non-private algorithms. Second, it needs the full knowledge of the computational graph due to backpropagation, while any insertion of black-box modules in the pipeline would block the use of DP-SGD. Third, it needs all operations to be differentiable, while many advanced models use non-differentiable activations, such as spiking neural networks (Tavanaei et al., 2019).

Different from deterministic backpropagation-based methods, forward-learning algorithms (Peng et al., 2022; Hinton, 2022; Salimans et al., 2017) employ perturbation or Monte Carlo simulations to estimate the gradient directly, bypassing the need for backpropagation based on the chain rule. Compared to backpropagation-based methods, forward-learning algorithms offer several advantages, including high parallelizability, suitability for non-differentiable modules, and applicability in black-box settings (Jiang et al., 2023). Moreover, as depicted in Figure 1, perturbation during the forward

pass naturally breaks the deterministic optimizations and results in randomized converged parameters, providing a potential "free lunch" for equipping the model with DP. Consequently, an intuitive question arises: *How could we utilize the inherent randomness in forward-learning algorithms to achieve differential privacy?*

To answer this question, we investigate the state-of-the-art forward-learning algorithm, the Unified Likelihood Ratio (ULR) method (Jiang et al., 2023). The ULR algorithm adds noise to intermediate values, e.g., each layer logit, during the forward propagation and utilizes theoretical tools to estimate the parameter gradients. While ULR inherently provides randomized gradients, there is still a gap in fully achieving differentially private learning.

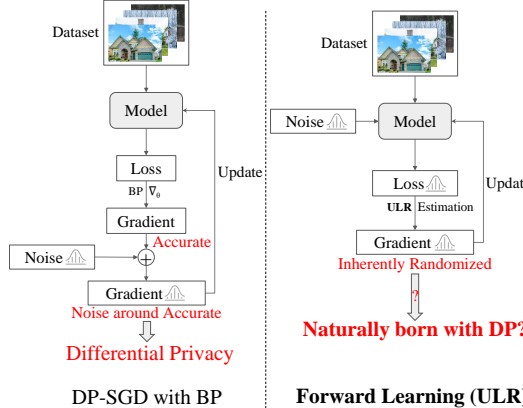

In this paper, to address this gap, we propose a privatized forward-learning algorithm, Differentially Private Unified Likelihood Ratio (DP-ULR), and provide a theoretical analysis of its differential privacy guarantees. DP-ULR distinguishes itself from ULR and achieves DP by incorporating novel elements, including the sampling-with-rejection strategy and the theoret-

Figure 1: Compared to traditional training algorithms, forward-learning adds noise during forward and estimates naturally randomized gradients, leading to a potential free lunch of differential privacy.

ically guided differential privacy controller. Although not treating DP-ULR as a drop-in replacement for DP-SGD, our theoretical analysis and experimental findings reveal that DP-ULR demonstrates nearly the same differential privacy properties and competitive utility in practice compared to DP-SGD. Our contributions are summarized as follows:

- We propose a novel sampling-with-rejection technique and theoretically analyze its impact on differential privacy in conjunction with the Gaussian mechanism.
- We introduce DP-ULR, a forward-learning differential privacy algorithm that integrates our sampling-with-rejection strategy and a well-designed differential privacy cost controller.
- We provide a comprehensive theoretical analysis of the differential privacy guarantees of our DP-ULR algorithm, establishing a general DP bound under typical conditions.
- We validate the effectiveness of our algorithm with MLP and CNN models on the MNIST and CIFAR-10 datasets.

## 2 BACKGROUND AND RELATED WORK

### 2.1 DIFFERENTIAL PRIVACY

Differential privacy (Dwork, 2006; Dwork et al., 2006b;a) is the gold standard for data privacy in controlling the disclosure of individual information. It is formally defined as the following:

**Definition 1** (($\epsilon, \delta$)-DP (Dwork et al., 2006a)). *A randomized mechanism $\mathcal{M}: \mathcal{D} \to \mathcal{R}$ with domain $\mathcal{D}$ and range $\mathcal{R}$, satisfies ($\epsilon, \delta$)-differential privacy if for any adjacent inputs $D, D' \in \mathcal{D}$ and for any subset of outputs $S \subseteq \mathcal{R}$ it holds that*

$$Pr[\mathcal{M}(D) \in S] \le e^{\epsilon} Pr[\mathcal{M}(D') \in S] + \delta. \tag{1}$$

The traditional privacy analysis of existing learning algorithms is obtained through Rényi Differential Privacy (Mironov, 2017; Mironov et al., 2019), which is defined with Rényi divergence.

**Definition 2** (Rényi divergence (Mironov, 2017; Mironov et al., 2019)). *Let $P$ and $Q$ be two distributions (random variables) defined over the same probability space, and let $p$ and $q$ be their respective probability density functions. The Rényi divergence of a finite order $\alpha \ne 1$ between $P$ and $Q$ is defined as*

$$D_{\alpha}(P \parallel Q) \coloneqq \frac{1}{\alpha - 1} \ln \mathbb{E}_{x \sim q} \left( \frac{p(x)}{q(x)} \right)^{\alpha} \tag{2}$$

*Rényi divergence at orders $\alpha = 1, \infty$ are defined by continuity.*

**Definition 3** (Rényi differential privacy (RDP) (Mironov, 2017; Mironov et al., 2019))**.** *We say that a randomized mechanism $\mathcal{M}\colon \mathcal{D} \to \mathcal{R}$ satisfies $(\alpha, \gamma)$-Rényi differential privacy (RDP) if for any two adjacent inputs $D, D' \in \mathcal{D}$ it holds that*

$$D_\alpha(\mathcal{M}(D) \parallel \mathcal{M}(D')) \le \gamma. \tag{3}$$

In this work, we use RDP to track privacy because of its outstanding composition property. Specifically, a sequence of $(\alpha, \gamma_i)$-RDP algorithms satisfies an additive RDP with $(\alpha, \sum_i \gamma_i)$. Moreover, we have the following proposition serving as a tool to transform the $(\alpha, \gamma)$-RDP to traditional $(\epsilon, \delta)$-DP.

**Proposition 1** (From $(\alpha, \gamma)$-RDP to $(\epsilon, \delta)$-DP (Mironov, 2017))**.** *If $f$ is an $(\alpha, \gamma)$-RDP mechanism, it also satisfies $(\gamma + \frac{\ln 1/\delta}{\alpha - 1}, \delta)$-differential privacy for any $0 < \delta < 1$, or equivalently $(\epsilon, \exp[(\alpha - 1)(\gamma - \epsilon)])$-differential privacy for any $\epsilon > \gamma$.*

## 2.2 DIFFERENTIAL PRIVACY IN DEEP LEARNING

As an adaption of Stochastic Gradient Descent (SGD) with backpropagation, DP-SGD (Abadi et al., 2016) is the most popular DP algorithm for deep learning (De et al., 2022; Sander et al., 2023). It assesses sensitivity by clipping the per-sample gradients and adds Gaussian noise after gradient computation to provide differential privacy guarantees. Particularly, as a training algorithm that comprises a sequence of adaptive mechanisms—a common scenario in deep learning—DP-SGD adds noise to the outcome of each sub-mechanism calibrated to its sensitivity, enhancing the utility of final learned models. While several techniques to improve the utility-privacy trade-off have been employed, including over-parameterized model (De et al., 2022), mega-batch training (Dörmann et al., 2021; Sander et al., 2023), averaging per-sample gradients across multiple augmentations (Hoffer et al., 2020), temporal parameter averaging (Polyak & Juditsky, 1992), equivariant networks (Hölzl et al., 2023), these adaptations not only heavily increase the computation cost but also do not change the core of DP-SGD: adding noise to deterministic gradients, which does not stand in forward learning.

## 2.3 FORWARD LEARNING

While there is no evidence that backpropagation exists in natural intelligence (Lillicrap et al., 2020), some studies put efforts into designing biologically plausible forward-only learning algorithms. For example, Nøkland (2016) employs the direct feedback alignment to train hidden layers independently. Jacot et al. (2018) leverage a neural target kernel to approximate the gradient for optimization. Salimans et al. (2017) apply the evolutional strategy to update the neural network parameters. Hinton (2022) replace the forward-backward pass with two forwards and optimize the neural networks by optimizing the local loss functions on positive and negative samples. Peng et al. (2022) propose a likelihood ratio (LR) method for unbiased gradient estimation with only one forward in the multi-layer perception training and Jiang et al. (2023) develop the unified likelihood ratio (ULR) method for training a wide range of deep learning models. In our work, we incorporate novel elements into ULR and provide a theoretical-guaranteed privatized forward-learning algorithm, *i.e.*, DP-ULR, to achieve differential privacy. We note that while several existing works (Liu et al., 2024b; Zhang et al., 2024; Tang et al., 2024) privatize loss values obtained in zeroth-order optimization for achieving DP, our work differs from them in multiple aspects, including motivation, application scope, the core algorithm, and theoretical analysis. Detailed discussion is provided in Appendix A.5.

## 3 METHODOLOGY

In Section 3.1, we present preliminaries of differential privacy in the deep learning setting. In Section 3.2, we introduce our proposed algorithm, DP-ULR. In Section 3.3, we provide our theoretical results of the DP bound. In Section 3.4, we discuss the difference between DP-ULR and DP-SGD.

### 3.1 PRELIMINARIES

In this paper, we consider the deep learning setting. Specifically, assume we have a (training) dataset $D = \{d_i\}_1^N$, where each example $d = (x, y) \in \mathcal{X} \times \mathcal{Y}$ is a pair of the input and the corresponding label. For a given model with a non-parameter structure $\varphi$ and a loss function $\ell$, the goal is to optimize the parameter $\theta$ to minimize the empirical loss, formalized as $\arg\min_\theta (\sum_{(x,y) \in D} \ell(\varphi(x; \theta), y))$.

Intuitively, the final output $\theta$ carries information from all examples as they all contribute to this optimization. In practice, deep-learning models do easily memorize sensitive, personal, or private data. For evaluating privacy in deep learning, differential privacy has become a significant criterion.

In the context of deep learning with differential privacy, a mechanism $\mathcal{M}$ refers to a training algorithm that takes a (training) dataset $D$, typically large, as the input and outputs a final parameter $\theta$. Thus, we consider the domain $\mathcal{D} = \{D \in 2^{\bar{D}} \mid |D| \geq \bar{N}\}$, where $\bar{N}$ is a positive integer and $\bar{D}$ is a large data pool, and the range $\mathcal{R} = \mathbb{R}^{d_\theta}$, where $d_\theta$ is the number of dimensions of the model parameter. Then, the adjacent inputs represent two training datasets $D, D' \in \mathcal{D}$ that differ by exactly one example. To guarantee privacy, we expect a randomized training algorithm $\mathcal{M}$ to produce effectively close final parameter distributions in terms of $(\epsilon, \delta)$-DP or $(\alpha, \gamma)$-RDP (Definition 1 and 3). For clarity, a complete list of symbols used in this paper can be found in Appendix A.1.

### 3.2 DP-ULR ALGORITHM

Consider a model with a hierarchical non-parameter structure $\varphi$ that can be sliced into $L$ modules. Let $\varphi^l$ denote the $l$-th module and $\theta^l$ denote the parameter of $\varphi^l$. We write $x^l$ for the $l$-th module's input and $v^l$ for the output, i.e., $v^l := \varphi^l(x^l; \theta^l)$. The outline of our DP-ULR training algorithm is depicted in Algorithm 1. Initially, in each step $t \in [T]$, we take an independent random sample from the dataset $D$ with equal sampling probability for each example. If the size of sampled batch $B_t$ is smaller than a pre-defined hyperparameter $N_B$, it is resampled. Subsequently, during the forward pass of each example $d = (x, y) \in B_t$, we inject Gaussian noise $z$ into each module's output $v^l$ separately. This noise-added output serves as the next module's input, i.e., $x^{l+1} = v^l + z$. Then, we compute the likelihood ratio gradient proxy $\hat{g}_t^l(d)$, which we define later in Equation (4). For each example $d$, we repeat $K$ times and clip the $\ell_2$ norm of averaged proxies to form the final estimated gradient $g_t^l(d)$. Finally, we employ the estimated gradients over the batch to update the parameter of the $l$-th module. During the whole process of training, we utilize a proxy controller method to adjust the standard deviation (std) $\sigma$ of noise given a required lower bound of the proxy's std $\sigma_0$ for the desire of differential privacy and to compute the accumulated privacy cost.

---

**Algorithm 1** Differential Private Likelihood Ratio Method (DP-ULR)

---

**Input:** Dataset $D = \{(x_i, y_i)\}_1^N$, loss function $\ell$, model structure $\varphi$. Parameters: learning rate $\eta_t$, target std $\sigma_0$, sampling rate $q$, rejection threshold $N_B$, repeat time $K$, overall clip bound $C$.

    **Initialize** $\theta_0$ randomly

    **for** $t \in [T]$ **do**

        Take a random sample $B_t$ from $D$ with sampling probability $N_B/N$, resample if $|B_t| < N_B$

        *// Estimate gradients*

        **for** $l \in L$ **do**

            Compute required noise std $\sigma$ (eq. (8)) and accumulate privacy cost using proxy controller

            **for** $d_i = (x_i, y_i) \in B_t$ **do**

                Sample $K$ zero-mean Gaussian noise $\{z_k\}_1^K \overset{\text{iid}}{\sim} \mathcal{N}(0, \sigma^2\mathbb{I})$

                Add noise to the $l$-th module's output $x_{i,k}^{l+1} = v_i^l + z_k = \varphi^l(x_i^l, \theta_t^l) + z_k, k = 1, ..., K$

                Forward to compute loss $\mathcal{L}_k = \ell(\varphi(x_i; \theta, l, z_k), y_i), k = 1, ..., K$

                Compute $\hat{g}_{t,k}^l(d_i) \leftarrow \frac{1}{\sigma^2} D_{\theta^l}^\top v_i^l \cdot z_k \mathcal{L}_k, k = 1, ..., K$     $\triangleright$ $D_{\theta^l}^\top v_i^l$ is the Jacobian matrix

                Compute $g_t^l(d_i) \leftarrow \frac{1}{K} \sum_k \hat{g}_{t,k}^l(d_i)$

                Clip gradient $g_t^l(d_i) \leftarrow g_t^l(d_i)/\max(1, \frac{\|g_t^l(d_i)\|_2}{C})$

        *// Gradient descent*

        For each layer $l$, $\theta_{t+1}^l \leftarrow \theta_t^l - \frac{\eta_t}{N_B} \sum_{d_i \in B_t} g_t^l(d_i)$

**Output:** $\theta_T = (\theta_T^1, ..., \theta_T^L)$ and the overall privacy cost

---

**Likelihood ratio proxy**. In our DP-ULR, we harness the likelihood ratio gradient proxy $\hat{g}^l(d)$ to approximate the ground-truth gradient for each example instead of accurately computing it by backpropagation. Let $\varphi(x; \theta, l, z)$ denote the model's output when the Gaussian noise $z \sim \mathcal{N}(0, \sigma^2\mathbb{I})$ is added to $v^l$. Then, the likelihood ratio gradient proxy before clipping is defined as

$$\hat{g}^l(d) = \frac{1}{\sigma^2}(D_{\theta^l} v^l)^\top \cdot z\mathcal{L}, \tag{4}$$

where $D_{\theta^l} v^l$ is the Jacobian matrix of $v^l$ with respect to $\theta^l$ and $\mathcal{L} := \ell(\varphi(x; \theta, l, z), y)$ represents the final noisy loss. A result from Jiang et al. (2023) detailed as Theorem 3 in the Appendix demonstrates that the expectation of our likelihood ratio gradient proxy equals the expectation of gradient with noise added, i.e., $\mathbb{E}_z(\hat{g}^l(d)) = \mathbb{E}_z(\nabla_{\theta^l} \mathcal{L})$. Subsequently, it follows with Proposition 2 below. It indicates that while the proxy leads to a certain precision loss in the gradient estimation, we can control it by selecting noise with a distribution close to 0, substantiating the utility of DP-ULR.

**Proposition 2.** *As the standard deviation $\sigma$ of noise approaches zero, the expectation of $\hat{g}^l(d)$ converges to the true gradient without noise, i.e.,*

$$\lim_{\sigma \downarrow 0} \mathbb{E}_z(\hat{g}^l(d)) = \nabla_{\theta^l} \ell(\varphi(x; \theta), y). \tag{5}$$

In addition to the expectation, we are also concerned about the proxy's variance from both perspectives of utility and privacy. Specifically, one intuitive question is how the variance of $g_t^l(d)$ changes as the std $\sigma$ of injected noise changes. Through asymptotic analysis, we show that the variance of the gradient proxy is inversely proportional to noise variance $\sigma^2$ when $\sigma$ is relatively small. Concretely, we state the following Proposition 3. The detailed analysis can be found in the Appendix A.3.

**Proposition 3.** *Given the loss without injected noise, $\mathcal{L}_0 := l(\varphi(x; \theta), y)$, and a small $\sigma$, we have*

$$\mathrm{Var}(\hat{g}^l(d)) \approx \frac{\mathcal{L}_0^2}{\sigma^2} (D_{\theta^l} v^l)^\top \cdot D_{\theta^l} v^l. \tag{6}$$

**Random distribution of estimated gradients**. Differential privacy guarantees are highly sensitive to the distribution of the mechanism's outputs. In the common strategy to protect privacy, the noise of a Gaussian distribution is added to the output, making it also a Gaussian distribution given the sampling result. On the contrary, in our method, the Gaussian noise is injected into the intermediate value, leaving the final output's distribution a mystery. Recall that the final gradient estimator is obtained by averaging $K$ repetitions. It follows that, according to the *multidimensional central limit theorem*, $g_t^l(d)$ can be approximated as Gaussian when the repeat time $K$ is large enough.

**Batch subsampling with rejection**. A significant difference between our proposed DP-ULR and the previous likelihood ratio methods is the subsampling operation. In DP-ULR, we adopt an independent sampling strategy with a predefined threshold $N_B$. Concretely, each example in the dataset $d_i \in D$ is picked independently with the same probability $q$. But if the size of $B_t$ is smaller than $N_B$, it is rejected and resampled. Like the ordinary i.i.d subsampling, our rejection strategy with a lower limit also amplifies privacy. Specifically, in the subsampling with rejection, we expect that the privacy cost $\gamma$ diminishes quadratically with the subsampling rate but adds a very small term that is not related to $\alpha$ in $(\alpha, \gamma)$-RDP. We discuss the privacy amplification in detail in Section 3.3.

In the implementation, batches are constructed by randomly permuting examples and then partitioning them into groups of fixed size for efficiency. For ease of analysis, however, we assume that each batch is formed by independently picking each example with the same probability $q$ and with rejection.

The intuition behind rejection is that, unlike traditional DP algorithms that add noise to the gradient with fixed variance independent of batch size, the variance of the gradient estimated by our method is directly and positively correlated with batch size. By rejecting small batches, we prevent the privacy costs of low randomness in extreme cases. Besides, in the setting of deep neural networks, the dimensions of the $l$-th module's parameter might be less than the dimensions of $l$-th module's output, i.e., $d_{\theta^l} > d_{v^l}$. Consequently, the Jacobian matrix $(D_{\theta^l} v^l)^\top$ would be a singular transformation of the high-dimensional random variable $z\mathcal{L}$. Then, the gradient proxy's covariance matrix must not be full-rank. Rank-deficient covariance is a dangerous signal in differential privacy because it means that the randomness is totally lost along certain directions in the high-dimensional space. In the quest to address this crisis, we introduce the following assumption.

**Assumption 1.** *There exists a positive integer $N_0$ less than $\bar{N}$, such that the sum of the covariance matrices of gradient proxies for any module and any batch with a size larger than $N_0$ sampled from any dataset is full rank. It is equivalent as follows, where we define $\Sigma_{\hat{g}(d)} := \mathrm{Var}(\hat{g}^l(d))$,*

$$\exists N_0 \in [\bar{N}], s.t. \forall D \in \mathcal{D}, \forall B \subset D, |B| \geq N_0, \forall l \in [L], rank(\sum_{d \in B} \Sigma_{\hat{g}(d)}) = d_{\theta^l}. \tag{7}$$

Assumption 1 indicates that setting the lower limit $N_B$ large enough provides a minimum guarantee of the output's randomness, enabling us to bound the privacy cost by a privacy controller. In Appendix A.4, we discuss when we expect Assumption 1 to hold.

**Privacy controller**. In our DP-ULR, we adopt a privacy-controlling method to guarantee the differential privacy cost for each step and, thus, the overall cost. The objective is to bound the minimum variance of the output, the estimated gradient in each step. Let $\Sigma_B$ denote the covariance matrix of the estimated gradient for the sampled batch $B$ and $\boldsymbol{\lambda}(\cdot)$ denote the spectrum of a matrix, i.e., the set of its eigenvalues. Note that we introduce Assumption 1 to ensure a full-rank covariance matrix if we set $N_B \geq N_0$. Equivalently, we have $\min(\boldsymbol{\lambda}(\Sigma_B)) > 0$. Then, Proposition 3 shows the feasibility of controlling $\min(\boldsymbol{\lambda}(\Sigma_B))$ by adjusting the std $\sigma$ of injected noise. Meanwhile, the adjustment must be dynamic because of the example-specific Jacobian matrix and non-noise loss.

Concretely, before computing likelihood ratio proxies in each step, we first execute one forward pass without any noise to obtain the Jacobian matrix $D_{\theta^l} v^l$ and the non-noisy loss $\mathcal{L}_0$ for each example. Subsequently, we compute the standard covariance matrix, $\tilde{\Sigma}_{\hat{g}(d)} := \mathcal{L}_0^2 (D_{\theta^l} v^l)^\top \cdot D_{\theta^l} v^l$, in the batch and the summation's minimum eigenvalue. Finally, we select suitable noise std $\sigma$ to bound the minimum eigenvalue of the covariance matrix of the estimated gradient. Mathematically, it requires

$$\sigma^2 \leq \frac{\min(\boldsymbol{\lambda}(\sum_{d \in B_t} \tilde{\Sigma}_{\hat{g}(d)}))}{K C^2 \sigma_0^2}, \tag{8}$$

where $\sigma_0$ is a predefined target std of estimated gradients. In the pseudocode of Algorithm 1, parameters are set as a constant. However, the independence of layers and steps allows for setting different target std scales $\sigma_0$, repeat time $K$, clipping thresholds $C$, and rejection thresholds $N_B$. For ease of the following analysis of differential privacy, we assume constant parameters at all times.

**Generalization of DP-ULR**. Our theoretical analysis focuses on the most general case of DP-ULR, highlighting its robustness and versatility. The DP-ULR method is highly adaptable; by considering different definitions of modules, we can adjust where noise is added, resulting in different variants of DP-ULR. Our theoretical framework generalizes well to these special cases. For instance, consider a virtual linear with the input of an identical matrix and the weight of the model parameters. Then, adding noise to the logit of this virtual linear layer equals adding noise to the model parameters directly, and the Jacobian matrix would be the identity matrix $\mathbb{I}$, ensuring the full rank.

**Remediation for violation of Assumption 1**. Changing where noise is added offers a remediation method if Assumption 1 is not satisfied under the standard module definition. Then, we can still ensure the privacy cost is controlled, albeit with some trade-off in network learning utility. Alternatively, extra independent noise can be added to the estimated gradient directly along its eigenvector directions, compensating for randomness deficiencies. We provide more details in Appendix A.4.

## 3.3 DIFFERENTIAL PRIVACY OF DP-ULR

In this section, we provide a theoretical analysis of the differential privacy of our DP-ULR algorithm. Let's first consider our subsampling operation with rejection in a typical Gaussian mechanism.

**Definition 4** (Sampled with Rejection Gaussian Mechanism (SRGM)). *Let $\mathcal{D}$ be a set of datasets. Assume datasets in $\mathcal{D}$ has a minimum size, i.e., $\exists \bar{N} > 1$, s.t. $\forall D \in \mathcal{D}$, $|D| > \bar{N}$. Let $f$ be a function mapping subsets of datasets in $\mathcal{D}$ to $\mathbb{R}^d$. We define the Sampled with Rejection Gaussian mechanism (SRGM) parameterized with the sampling rate $0 < q \leq 1$, the noise $\sigma > 0$, and the lower limit $1 \leq N_B \leq \bar{N}$ as*

$$SRG_{q,\sigma,N_B}(D) := f(\bar{D}) + \mathcal{N}(0, \sigma^2 \mathbb{I}^d), \tag{9}$$

*where each element of $D$ is sampled independently at random with probability $q$ without replacement to form $\bar{D}$, $\bar{D}$ is rejected and resampled if $|\bar{D}| < N_B$, and $\mathcal{N}(0, \sigma^2 \mathbb{I}^d)$ is spherical $d$-dimensional Gaussian noise with per-coordinate variance $\sigma^2$.*

SRGM is similar to the well-studied Sampled Gaussian mechanism (SGM), whose privacy bound has been derived in several settings(Mironov et al., 2019). Since with a new parameter, rejection threshold $N_B$, the SRGM requires a lower bound of the dataset size in the domain $\mathcal{S}$ and has a different impact on the differential privacy. Based on previous studies, we introduce the following theorem.

**Theorem 1.** *If $1 \leq N_B \leq q\bar{N}$, $q \leq \frac{1}{5}$, $\sigma \geq 4$, and $\alpha$ satisfy $1 < \alpha \leq \frac{1}{2}\sigma^2 A - 2\ln\sigma$ and $\alpha \leq \frac{\frac{1}{2}\sigma^2 A^2 - \ln 5 - 2\ln\sigma}{A + \ln(q\alpha) + 1/(2\sigma^2)}$, where $A := \ln\left(1 + \frac{1}{q(\alpha-1)}\right)$, then SRGM applied to a function of $\ell_2$-sensitivity 1 satisfies $(\alpha, \gamma)$-RDP for*

$$\gamma = \frac{q\boldsymbol{p}(N_B - 1; \bar{N}, q)}{1 - \boldsymbol{P}(N_B - 1; \bar{N}, q)} + \frac{2q^2}{\sigma^2}\alpha, \tag{10}$$

*where $\boldsymbol{p}(\cdot, \bar{N}, q)$ and $\boldsymbol{P}(\cdot, \bar{N}, q)$ are defined as the probability mass function and cumulative distribution function of the binomial distribution with parameters $\bar{N}$ and $q$, respectively.*

Theorem 1 indicates a quadratic amplification with the subsampling rate $q$ but also a certain impairment on the privacy cost of SRGM. But in our case, the distribution of the estimated gradient is not isotropic. Particularly, its covariance matrix is varying rather than constant and irrelevant to the sampled batch in each step, making it difficult to derive a universal bound. However, conditioned on Assumption 1, we utilize the privacy controller to guarantee a minimum level of randomness $\sigma_0^2$. Then, we state our main theorem about DP-ULR below. The proof can be found in Appendix B.

**Theorem 2.** *Assume $\sigma^2$ satisfies Equation (8). Then, if $1 \le N_B \le q\bar{N}$, $q \le \frac{1}{5}$, $\sigma_0 \ge 4$, and $\alpha$ satisfy $1 < \alpha \le \frac{1}{2}\sigma_0^2 A - 2\ln\sigma_0$ and $\alpha \le \frac{\frac{1}{2}\sigma_0^2 A^2 - \ln 5 - 2\ln\sigma_0}{A + \ln(q\alpha) + 1/(2\sigma_0^2)}$, where $A := \ln\left(1 + \frac{1}{q(\alpha-1)}\right)$, DP-ULR (Algorithm 1) satisfies $(\alpha, \gamma)$-RDP for*

$$\gamma = \frac{Tq\boldsymbol{p}(N_B - 1; \bar{N}, q)}{1 - \boldsymbol{P}(N_B - 1; \bar{N}, q)} + \frac{2Tq^2}{\sigma_0^2}\alpha, \tag{11}$$

*where $\boldsymbol{p}(\cdot, \bar{N}, q)$ and $\boldsymbol{P}(\cdot, \bar{N}, q)$ are defined as the probability mass function and cumulative distribution function of the binomial distribution with parameters $\bar{N}$ and $q$, respectively.*

### 3.4 DP-SGD *v.s.* DP-ULR

In this section, we discuss the difference between DP-SGD and DP-ULR.

**Minute DP impairment**. According to Mironov et al. (2019), DP-SGD(Abadi et al., 2016) satisfies $(\alpha, \gamma)$-RDP for a suitable range of $\alpha$ and $\gamma = 2Tq^2\alpha/\sigma^2$, where $\sigma$ is noise scale. If we set our target output std $\sigma_0$ equal to this noise scale, the difference in RDP bound is the impairment term from the SRGM, which is related to the training dataset size. In practice, deep learning datasets are quite large. Subsequently, the impairment term is extremely small to be ignored compared to the latter term. For instance, if we consider $\bar{N} = 10000$, $q = 0.01$, and $N_B = 50$, the impairment is less than $10^{-10}$, while the second term is greater than $10^{-6}$. We provide further empirical analysis in Section 4.1.

**Noise redundancy**. DP-SGD injects isotropic noise directly into the precise gradient, making full use of noise to offer differential privacy. DP-ULR attempts to utilize the inherent randomness of gradient estimation, where noise is added to the intermediate values in the forward pass. It provides privacy protection by ensuring the variance of the estimated gradient in an arbitrary direction no less than a pre-defined level. However, it also means that randomness in many other directions is even larger than this level due to the non-isotropy. This redundant noise doesn't contribute to the bound of differential privacy but impairs the accuracy of the gradient estimation.

**Efficiency and suitability**. A common limitation of DP-SGD is its slower speed compared to traditional SGD, primarily due to the requirement to clip each individual gradient, necessitating an independent backward pass for each example. In contrast, the computation of individual estimated gradients in DP-ULR is inherently separate, allowing for individual clipping without additional computational cost compared to ULR. Additionally, as a variant of ULR, DP-ULR inherits certain advantages over backpropagation-based DP-SGD, including suitability for non-differentiable or black-box settings, high parallelizability, and efficient pipeline design (Jiang et al., 2023). Furthermore, in cases where the loss function cannot be expressed as a summation of individual losses, computing individual gradients to limit example sensitivity becomes challenging for standard backpropagation. In DP-ULR, noise can be injected separately, enabling independent gradient computation, thus broadening its potential applications.

## 4 EXPERIMENTS

### 4.1 ANALYSIS OF DP BOUND

In our approach, the introduction of the sampling-with-rejection technique ensures adequate randomness across all directions in the parameter space at each step of training. As detailed in Equation 11, the sampling-with-rejection operation introduces an additional term to the DP cost, not present in traditional algorithms that employ common i.i.d. Poisson sampling. Despite this, we illustrate that the impact is minimal in typical deep-learning scenarios.

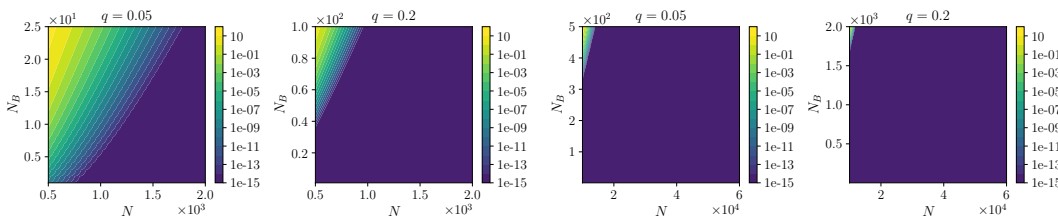

Figure 2: Contour plots of the ratio of the first term to the second term in Equation (11).

| Method | | **DP-ULR** | | **DP-SGD** | |
|---|---|---|---|---|---|
| Batch size | $\sigma_0$ | Training acc.(%) | Valid acc. (%) | Training acc.(%) | Valid acc. (%) |
| 64 | 0.5 | $85.43_{\pm0.53}$ | $86.12_{\pm0.45}$ | $93.94_{\pm0.15}$ | $94.14_{\pm0.18}$ |
| | 1 | $81.81_{\pm0.60}$ | $82.73_{\pm1.20}$ | $90.90_{\pm0.08}$ | $91.20_{\pm0.10}$ |
| | 2 | $78.04_{\pm0.94}$ | $78.80_{\pm1.42}$ | $78.61_{\pm0.80}$ | $78.32_{\pm0.84}$ |
| | 4 | $70.71_{\pm2.08}$ | $72.14_{\pm2.19}$ | $67.33_{\pm1.04}$ | $68.73_{\pm1.74}$ |
| | 8 | $57.88_{\pm4.53}$ | $58.65_{\pm6.32}$ | $33.63_{\pm0.65}$ | $32.07_{\pm4.17}$ |
| 200 | 0.5 | $91.55_{\pm0.19}$ | $91.98_{\pm0.25}$ | $90.47_{\pm0.25}$ | $90.68_{\pm0.30}$ |
| | 1 | $89.24_{\pm0.49}$ | $90.11_{\pm0.59}$ | $90.54_{\pm0.24}$ | $90.77_{\pm0.31}$ |
| | 2 | $86.15_{\pm0.41}$ | $87.11_{\pm0.59}$ | $90.63_{\pm0.14}$ | $90.95_{\pm0.29}$ |
| | 4 | $83.20_{\pm0.49}$ | $84.45_{\pm0.62}$ | $89.07_{\pm0.16}$ | $89.63_{\pm0.14}$ |
| | 8 | $78.91_{\pm1.12}$ | $80.05_{\pm0.50}$ | $78.50_{\pm0.77}$ | $79.14_{\pm0.67}$ |
| 500 | 0.5 | $93.92_{\pm0.15}$ | $94.19_{\pm0.15}$ | $87.56_{\pm0.59}$ | $87.98_{\pm0.64}$ |
| | 1 | $91.73_{\pm0.25}$ | $92.04_{\pm0.34}$ | $87.56_{\pm0.60}$ | $87.98_{\pm0.62}$ |
| | 2 | $89.33_{\pm0.42}$ | $90.31_{\pm0.55}$ | $87.59_{\pm0.60}$ | $88.00_{\pm0.62}$ |
| | 4 | $86.56_{\pm0.80}$ | $87.63_{\pm0.80}$ | $87.66_{\pm0.51}$ | $87.97_{\pm0.61}$ |
| | 8 | $82.87_{\pm0.89}$ | $84.53_{\pm0.71}$ | $87.68_{\pm0.47}$ | $88.16_{\pm0.59}$ |

Table 1: The classification accuracy of MLP on the MNIST dataset.

Figure 2 provides contour plots of the ratio between the first and second terms of the DP cost across various dataset sizes and rejection thresholds, based on theoretical results with parameters $\alpha = 1.1$ and $\sigma_0 = 4$. As shown, when the dataset size $N \geq 10^3$ and the rejection threshold $N_B$ is slightly less than the mean batch size $qN$ (if without rejection), the ratio of the first impairment term (introduced by rejection sampling) to the second term is less than $10^{-3}$. This empirical evidence suggests that the increased privacy costs due to our sampling method are effectively negligible.

Moreover, as the dataset size increases, the relative impact of the first term on the privacy cost diminishes further, underlining the suitability of our method for training on large-scale datasets. This scalability is crucial for deploying differential privacy in real-world applications where large models are trained on extensive data collections.

## 4.2 EVALUATIONS ON MLP

**Model and dataset**. **Model and Dataset:** We evaluated our proposed DP-ULR method by training a multilayer perceptron (MLP) with four layers containing $128$, $64$, $32$, and $10$ neurons, respectively, each employing GELU activations. The MLP was trained on the MNIST dataset, comprising $60,000$ training images and $10,000$ validation images across 10 classes.

**Experiment Settings**. We configured DP-ULR with a learning rate of 0.01, utilizing the Adam optimizer with cross-entropy loss. We utilize the approach of adding extra noise to remediate the violence of the full rank. We compare our method with DP-SGD, utilizing the OPACUS open-source implementation. For DP-SGD, we use the default settings: a learning rate of 0.1 with the SGD optimizer. We also experimented with the Adam optimizer and a learning rate of 0.01 but found the default settings provided better performance. Furthermore, we tested DP-SGD using both standard Poisson sampling, which is required by the theory, and a fixed batch size implementation, observing minimal performance differences. Thus, for consistency, results with the fixed batch size implementation are reported. For both DP-ULR and DP-SGD, the learning rate is reduced by 0.85 every 10 epoch, training is conducted over 25 epochs, and the clipping threshold $C$ is set to 1.

We fix the target $\delta = 10^{-5}$ and experiment with different settings of batch size $B = 64, 200, 500$, corresponding to different sample rates $q = 10^{-3}, \frac{1}{300}, \frac{1}{120}$, and target std level (noise level) $\sigma_0 =$

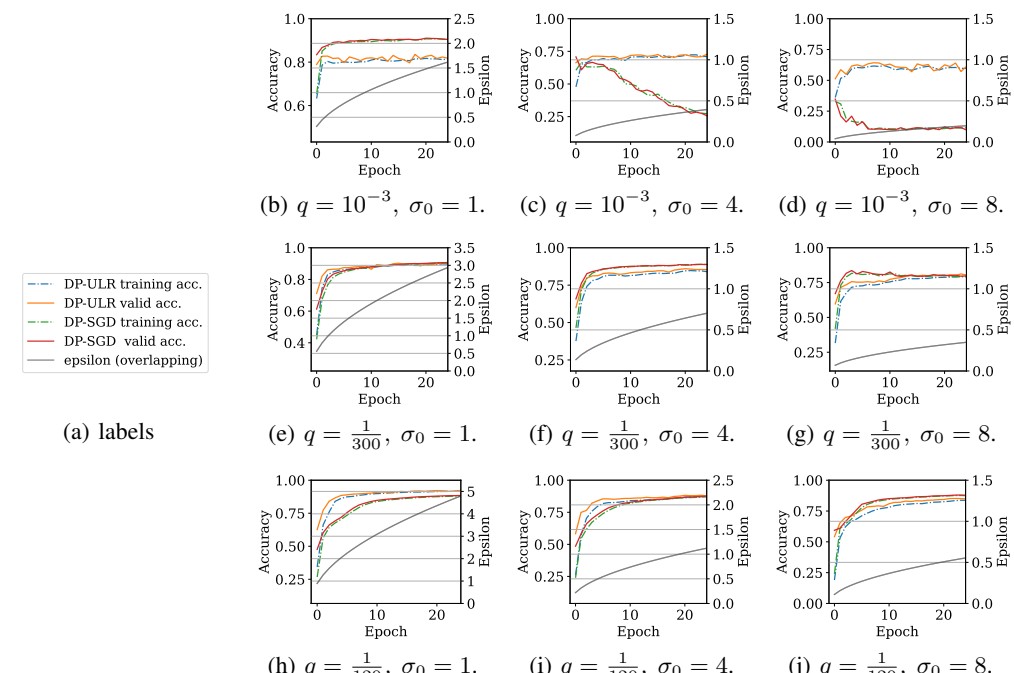

Figure 3: Optimization dynamics of the MLP training with differential privacy using DP-SGD and our proposed DP-ULR and corresponding $\epsilon$ with $\delta = 10^{-5}$.

$0.5, 1, 2, 4, 8$. For all settings, we repeat the experiments 5 times with different random seeds and report the average and standard deviations. We also conduct ablation experiments on model sizes, of which results and analysis are provided in Appendix C.1.

**Experiment Results**. Table 1 presents the training and validation accuracies for DP-ULR and DP-SGD across varying batch sizes and target std levels or noise levels. For DP-ULR, we report the final epoch accuracy, while for DP-SGD, due to potential severe performance degradation over iterations, we report the highest accuracy achieved during training if needed. We can see that DP-ULR shows improved performance with larger batch sizes, while DP-SGD performs better with smaller batch sizes. This is probably because the noise redundancy is more severe in smaller batch sizes, degrading the accuracy of gradient estimation. Consequently, DP-ULR outperforms DP-SGD with the large batch, underperforms with the small batch, and has a competitive performance compared to DP-SGD. Another interesting observation is that DP-ULR is more sensitive to noise scale $\sigma_0$ in large batch sizes, whereas DP-SGD shows greater sensitivity to $\sigma_0$ in small batch sizes.

Figure 3 illustrates the optimization dynamics of training and valid accuracy alongside the corresponding $\epsilon$ value with fixed $\delta = 10^{-5}$. We report the results with different sample rates $q = 10^{-3}, \frac{1}{300}, \frac{1}{120}$ and noise level $\sigma_0 = 1, 4, 8$. To present a fair comparison, we compute the $\epsilon$ of DP-SGD using the RDP bound from Mironov et al. (2019) rather than the value provided by OPACUS, which is computed differently. The overlapping curves of $\epsilon$ in Figure 3 suggest that the impairment term in the DP bound is negligible. Our results indicate that, under the same batch size and high noise levels, DP-SGD suffers from performance degradation, whereas DP-ULR continues to converge. Both methods exhibit minimal differences between training and validation accuracies, and in some instances, validation accuracy surpasses training accuracy, particularly in the early training stages.

### 4.3 EVALUATIONS ON CNN

We further evaluate the performance of DP-ULR by training a CNN on the CIFAR-10 dataset. The CIFAR-10 dataset has a training set of 50000 images and a test set of 10000 images. We use the ResNet-5 as our studied model. The ResNet-5 has 5 layers, including 4 convolutional layers and 1 fully connected layer. The residual connection is between the third and fourth convolutional layers. For the convolutional layers, we set the number of kernels as 8, 16, 32, and 32, respectively, and all the kernel sizes as $3 \times 3$ with the stride as 1 and the activation function as ReLU.

We test our DP-ULR with different sample rates $q$ and target std level $\sigma_0$. We experiment with DP-SGD setting batch size as $64$ and noise level as $1$. The results are shown in Figure 4. We can see that during the training with DP-ULR, the convergence of the model fluctuates and sometimes drops abruptly. Nevertheless, by selecting suitable parameters, our proposed DP-ULR can achieve comparable performance in the end with DP-SGD in terms of both accuracy and privacy cost.

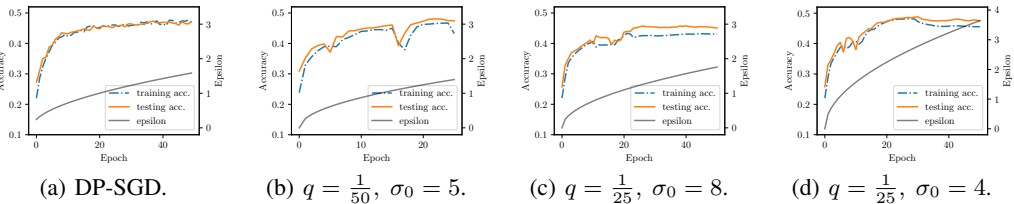

(a) DP-SGD.     (b) $q = \frac{1}{50}$, $\sigma_0 = 5$.     (c) $q = \frac{1}{25}$, $\sigma_0 = 8$.     (d) $q = \frac{1}{25}$, $\sigma_0 = 4$.

Figure 4: Evaluation results of the CNN training with differential privacy using DP-SGD in (a) and our proposed DP-ULR in (b)–(d).

## 5 CONCLUSIONS

In this paper, we propose a forward-learning DP algorithm, Differential Private Unified Likelihood Ratio (DP-ULR). Unlike traditional backpropagation-based methods such as DP-SGD, which rely on computing individual gradients and adding noise, DP-ULR leverages the inherent randomness in forward-learning algorithms to achieve differential privacy. Our approach introduces a novel batch sampling operation with rejection and a dynamically managed privacy controller to ensure robust privacy guarantees.

Our theoretical analysis demonstrates that the additional privacy cost introduced by the sampling-with-rejection operation is negligible, particularly in large-scale deep-learning applications. This indicates the scalability and efficiency of DP-ULR in practical settings. Furthermore, our empirical results show that DP-ULR performs competitively compared to traditional DP training algorithms, maintaining the same privacy loss constraints while offering high parallelizability and suitability for non-differentiable or black-box modules.

In summary, DP-ULR provides a promising alternative to existing differential privacy methods, combining the benefits of forward learning with rigorous privacy guarantees.

## LIMITATIONS AND FUTURE WORK

We proposed an intuitive and direct adaptation (DP-ULR) of a forward-learning approach (ULR) that diverges from traditional SGD by eschewing backpropagation. Our analysis in this work primarily compares DP-ULR with the canonical form of DP-SGD. We acknowledge that recent advancements that incrementally improve the *privacy-utility trade-off* in DP-SGD could potentially be generalized to our forward-learning context; however, such extensions are beyond the scope of our initial investigation and represent promising avenues for future research.

Although DP-ULR retains the same benefits as ULR due to the unchanged core mechanics, we did not explore its suitability for non-differential or black-box settings in our experiments. Additionally, we did not implement parallelization or optimize the training pipeline for efficiency. In its current implementation, DP-ULR takes 23 seconds per epoch on an A6000 GPU with the MNIST dataset, which is slower than DP-SGD, which takes 18 seconds per epoch. Due to the large gradient estimation variance, the scale-up of ULR usually requires a large number of copies, which poses a great challenge to the computation and memory cost. Future works might focus on the development of advanced techniques to improve computational efficiency and reduce the estimation variance.

Differential privacy aims to ensure data privacy through randomness. When used to train a deep learning model, such randomness impairs the model's performance. Our algorithm has the same limitation. Besides, during training with DP-ULR, we observed overfitting, where the model achieved high accuracy (around 90% on MNIST) but exhibited extreme losses: near-zero loss for some samples while having very high losses for others that it failed to classify. Addressing this overfitting issue is another area for potential future exploration.

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

# A DIFFERENTIALLY PRIVATE UNIFIED LIKELIHOOD RATIO METHOD

## A.1 LIST OF SYMBOLS

| | |
|---|---|
| $\mathcal{D}$ | domain (set of dataset) |
| $\bar{D}$ | large data pool |
| $D$ | dataset |

| | |
|---|---|
| $N$ | size of dataset (number of examples) |
| $\bar{N}$ | lower limit of the size of dataset |
| $N_B$ | rejection threshold (hyperparameter) |
| $N_0$ | a level of batch size mentioned in Assumption 1 |

| | |
|---|---|
| $B$ | batch (sample from dataset) |
| $d$ | example |
| $x$ | input |
| $y$ | label |
| $\varphi$ | non-parameter structure of model |
| $\theta$ | parameter of model |
| $d_\theta$ | number of dimensions of parameter |
| $v$ | output of model |
| $z$ | noise (random variable) |
| $\sigma$ | std of noise |
| $\sigma_0$ | a required level of the likelihood ratio proxy's std (hyperparameter) |
| $\Sigma$ | covariance matrix |

| | |
|---|---|
| $\eta$ | learning rate (hyperparameter) |
| $q$ | sampling rate (hyperparameter) |
| $K$ | repeat time (hyperparameter) |
| $C$ | overall clip bound (hyperparameter) |

| | |
|---|---|
| $\mathbb{E}$ | expectation |
| $f(\cdot)$ | probability density function |
| $l(\cdot, \cdot)$ | loss function (a function) |
| $\mathcal{L}$ | loss (a variable) |
| $\mathcal{L}_0$ | loss without injected noise |
| $L$ | number of layers |
| $T$ | number of training steps |
| $l$ | index of layer, $l = 1, ..., L$ |
| $t$ | index of training step, $l = 1, ..., T$ |
| $t$ | index of example |

| | |
|---|---|
| $x^l$ | input of $l$-th layer |
| $\varphi^l$ | non-parameter structure of $l$-th layer |
| $\theta^l$ | parameter of $l$-th layer |
| $v^l$ | output of $l$-th layer |
| $z^l$ | noise that we add to $v^l$, $l = 1, ..., L-1$ |
| $d_l$ | dimension of $x^l$ |

## A.2 RESTATEMENT OF THE PREVIOUS THEOREM

We restate the Theorem 1 from the previous work Jiang et al. (2023).

**Theorem 3.** *Given an input data $x$, assume that $g^l(\xi) := f(\xi - \varphi^l(x^l; \theta^l))$ is differentiable, and*

$$\mathbb{E}\left[\int_{\mathbb{R}^{d_{l+1}}} \left|\mathbb{E}\left[\mathcal{L}(x^L)|\xi, x^l\right]\right| \sup_{\theta^l \in \Theta^l} \left|\nabla_{\theta^l} g^l(\xi)\right| d\xi\right] < \infty. \tag{12}$$

*Then, we have*

$$\nabla_{\theta^l}\mathbb{E}\left[\mathcal{L}(x^L)\right] = \mathbb{E}\left[-\mathcal{L}(x^L)J_{\theta^l}^\top \varphi^l(x^l; \theta^l)\nabla_z \ln f^l(z^l)\right]. \tag{13}$$

*Proof.* To update the $l$-th layer's parameter, we need to calculate the gradient for $\theta^l$. We have

$$\nabla_{\theta^l}\mathbb{E}_{z^1,\dots,z^{L-1}}[\mathcal{L}(v^L,y)] = \nabla_{\theta^l}\mathbb{E}_{z^1,\dots,z^{l-1}}\left[\mathbb{E}_{z^l,\dots,z^{L-1}}[\mathcal{L}(v^L,y) \mid v^l]\right]$$

$$= \nabla_{\theta^l}\mathbb{E}_{z^1,\dots,z^{l-1}}\left[\mathbb{E}_{z^l}\left[\mathbb{E}_{z^{l+1},\dots,z^{L-1}}[\mathcal{L}(v^L,y) \mid z^l,v^l] \mid v^l\right]\right].$$

The conditional expectation $\mathbb{E}_{z^{l+1},\dots,z^{L-1}}[\mathcal{L}(v^L,y) \mid z^l,v^l]$ is only related to the sum of $v^l$ and $z^l$, $x^{l+1} := v^l + z^l$. It means $\mathbb{E}_{z^{l+1},\dots,z^{L-1}}[\mathcal{L}(v^L,y) \mid z^l,v^l] = \mathbb{E}_{z^{l+1},\dots,z^{L-1}}[\mathcal{L}(v^L,y) \mid x^{l+1}]|_{x^{l+1}=v^l+z^l}$. We denote $h(\zeta) := \mathbb{E}_{z^{l+1},\dots,z^{L-1}}[\mathcal{L}(v^L,y) \mid x^{l+1}]|_{x^{l+1}=\zeta}$. Then, we have

$$\nabla_{\theta^l}\mathbb{E}_{z^1,\dots,z^{L-1}}[\mathcal{L}(v^L,y)] = \nabla_{\theta^l}\mathbb{E}_{z^1,\dots,z^{l-1}}\left[\mathbb{E}_{z^l}[h(v^l+z^l) \mid v^l]\right]$$

$$= \nabla_{\theta^l}\mathbb{E}_{z^1,\dots,z^{l-1}}\left[\int_{\mathbb{R}^{d_{l+1}}} h(v^l+\zeta)f_{z^l}(\zeta)d\zeta\right]$$

By changing the variable $\zeta$ to $\xi = v^l + \zeta$, we have

$$\int_{\mathbb{R}^{d_{l+1}}} h(v^l+\zeta)f_{z^l}(\zeta)d\zeta = \int_{\mathbb{R}^{d_{l+1}}} h(\xi)f_{z^l}(\xi-v^l)d\xi. \tag{14}$$

Since $h(\cdot)$ is not related to $\theta^l$ and $v^l = \varphi^l(x^l;\theta^l)$, we have

$$\nabla_{\theta^l}\mathbb{E}_{z^1,\dots,z^{L-1}}[\mathcal{L}(v^L,y)] = \nabla_{\theta^l}\mathbb{E}_{z^1,\dots,z^{l-1}}\left[\int_{\mathbb{R}^{d_{l+1}}} h(\xi)f_{z^l}(\xi-v^l)d\xi\right]$$

$$= \mathbb{E}_{z^1,\dots,z^{l-1}}\left[\int_{\mathbb{R}^{d_{l+1}}} \nabla_{\theta^l}\left(h(\xi)f_{z^l}(\xi-v^l)\right)d\xi\right]$$

$$= \mathbb{E}_{z^1,\dots,z^{l-1}}\left[\int_{\mathbb{R}^{d_{l+1}}} h(\xi)\nabla_{\theta^l}f_{z^l}(\xi-v^l)d\xi\right].$$

By the chain rule,

$$\nabla_{\theta^l}f_{z^l}(\xi-v^l) = -D_{\theta^l}^\top v^l \cdot \nabla_\zeta f_{z^l}(\zeta)|_{\zeta=\xi-v^l}, \tag{15}$$

where $D_{\theta^l}v^l \in \mathbb{R}^{d_{l+1}\times d_{\theta^l}}$ is the Jacobian matrix of $v^l = \varphi^l(x^l;\theta^l)$ with respect to $\theta^l$. Thus, we have

$$\nabla_{\theta^l}\mathbb{E}_{z^1,\dots,z^{L-1}}[\mathcal{L}(v^L,y)] = \mathbb{E}_{z^1,\dots,z^{l-1}}\left[\int_{\mathbb{R}^{d_{l+1}}} \left(-h(\xi)D_{\theta^l}^\top v^l \cdot \nabla_\zeta f_{z^l}(\zeta)|_{\zeta=\xi-v^l}\right)d\xi\right] \tag{16}$$

By changing the variable from $\xi$ back to $\zeta = \xi - v^l$, we have

$$\nabla_{\theta^l}\mathbb{E}_{z^1,\dots,z^{L-1}}[\mathcal{L}(v^L,y)]$$

$$= \mathbb{E}_{z^1,\dots,z^{l-1}}\left[\int_{\mathbb{R}^{d_{l+1}}} \left(-h(\zeta+v^l)D_{\theta^l}^\top v^l \cdot \nabla_\zeta f_{z^l}(\zeta)\right)d\zeta\right]$$

$$= \mathbb{E}_{z^1,\dots,z^{l-1}}\left[\int_{\mathbb{R}^{d_{l+1}}} \left(-h(\zeta+v^l)D_{\theta^l}^\top v^l \cdot \nabla_\zeta \ln f_{z^l}(\zeta)\right)f_{z^l}(\zeta)d\zeta\right]$$

$$= \mathbb{E}_{z^1,\dots,z^{l-1}}\left[\mathbb{E}_{z^l}\left[-h(z^l+v^l)D_{\theta^l}^\top v^l \cdot \nabla_\zeta \ln f_{z^l}(\zeta)|_{\zeta=z^l} \mid v^l\right]\right]$$

$$= \mathbb{E}_{z^1,\dots,z^{l-1}}\left[\mathbb{E}_{z^l}\left[-\mathbb{E}_{z^{l+1},\dots,z^{L-1}}[\mathcal{L}(v^L,y) \mid z^l,v^l]D_{\theta^l}^\top v^l \cdot \nabla_\zeta \ln f_{z^l}(\zeta)|_{\zeta=z^l} \mid v^l\right]\right]$$

$$= \mathbb{E}_{z^1,\dots,z^{l-1}}\left[\mathbb{E}_{z^l}\left[\mathbb{E}_{z^{l+1},\dots,z^{L-1}}[-\mathcal{L}(v^L,y)D_{\theta^l}^\top v^l \cdot \nabla_\zeta \ln f_{z^l}(\zeta)|_{\zeta=z^l} \mid z^l,v^l] \mid v^l\right]\right]$$

$$= \mathbb{E}_{z^1,\dots,z^{l-1}}\left[\mathbb{E}_{z^l,\dots,z^{L-1}}\left[-\mathcal{L}(v^L,y)D_{\theta^l}^\top v^l \cdot \nabla_\zeta \ln f_{z^l}(\zeta)|_{\zeta=z^l} \mid v^l\right]\right]$$

$$= \mathbb{E}_{z^1,\dots,z^{L-1}}\left[-\mathcal{L}(v^L,y)D_{\theta^l}^\top \varphi^l(x^l;\theta^l) \cdot \nabla_\zeta \ln f_{z^l}(\zeta)|_{\zeta=z^l}\right].$$

$\square$

## A.3 DISTRIBUTION OF ESTIMATED GRADIENTS

For a specific example, $(x_i,y_i) \in \mathcal{D}$, suppose each sampling outcome $D_{\theta^l}^\top v_i^l \cdot \frac{1}{\sigma^2}z\mathcal{L}$ has the mean vector $\mu_t^{l,i}$ and the covariance matrix $\Sigma_t^{l,i}$. Then, the estimated gradient $g_t^l(x_i)$ can be seen as a multivariate Gaussian distribution, $\mathcal{N}(\mu_t^{l,i}, \frac{1}{K}\Sigma_t^{l,i})$, when $K$ is large enough, according to the multidimensional central limit theorem.

Using multivariate Taylor expansion, we can say

$$\mathcal{L} = \mathcal{L}_0 + \nabla_{x^{l+1}}\mathcal{L}|_{x^{l+1}=v^l} \cdot z + \frac{1}{2}\nabla^2_{x^{l+1}}\mathcal{L}|_{x^{l+1}=v^l} \cdot z^2 \tag{17}$$

Then we have,

$$\frac{1}{\sigma^2}z\mathcal{L} = \frac{1}{\sigma^2}(\mathcal{L}_0 + \nabla_{x^{l+1}}\mathcal{L}|_{x^{l+1}=v^l} \cdot z + \frac{1}{2}\nabla^2_{x^{l+1}}\mathcal{L}|_{x^{l+1}=v^l} \cdot z^2)z \tag{18}$$

Then the expectation of $\frac{1}{\sigma^2}z\mathcal{L}$ is

$$\mathbb{E}(\frac{1}{\sigma^2}z\mathcal{L}) = 0 + \frac{1}{\sigma^2}\nabla_{x^{l+1}}\mathcal{L}|_{x^{l+1}=v^l} \cdot \sigma^2\mathbb{I} + 0 = \nabla_{x^{l+1}}\mathcal{L}|_{x^{l+1}=v^l} \tag{19}$$

Then the covariance matrix of $\frac{1}{\sigma}z_k\mathcal{L}_k$ is

$$\mathrm{Var}(\frac{1}{\sigma^2}z\mathcal{L}) = \mathrm{Cov}(\frac{1}{\sigma}z\mathcal{L}, \frac{1}{\sigma^2}z\mathcal{L}) \tag{20}$$

$$= \mathbb{E}\left(\frac{\mathcal{L}^2}{\sigma^4}z \cdot z^\top\right) - \nabla_{x^{l+1}}\mathcal{L}|_{x^{l+1}=v^l} \cdot \nabla^\top_{x^{l+1}}\mathcal{L}|_{x^{l+1}=v^l} \tag{21}$$

$$= \frac{1}{\sigma^2}\mathcal{L}_0\mathbb{I} + (2\mathcal{L}_0\nabla^2_{x^{l+1}}\mathcal{L}|_{x^{l+1}=v^l} + \nabla_{x^{l+1}}\mathcal{L}|_{x^{l+1}=v^l} \cdot \nabla^\top_{x^{l+1}}\mathcal{L}|_{x^{l+1}=v^l} \tag{22}$$

$$+ \mathrm{tr}(\mathcal{L}_0\nabla^2_{x^{l+1}}\mathcal{L}|_{x^{l+1}=v^l} + \nabla_{x^{l+1}}\mathcal{L}|_{x^{l+1}=v^l} \cdot \nabla^\top_{x^{l+1}}\mathcal{L}|_{x^{l+1}=v^l})\mathbb{I}) + \sigma^2... \tag{23}$$

When $\sigma$ approaches zero, the first term dominates others. Therefore, we have

$$\mathrm{Var}(\frac{1}{\sigma^2}z\mathcal{L}) \approx \frac{\mathcal{L}_0^2}{\sigma^2}\mathbb{I}. \tag{24}$$

Since $\hat{g}^l(d) = \frac{1}{\sigma^2}D_{\theta^l}^\top v^l \cdot z\mathcal{L}$, we have

$$\mathrm{Var}(\hat{g}^l(d)) \approx \frac{\mathcal{L}_0^2}{\sigma^2}D_{\theta^l}^\top v^l \cdot D_{\theta^l}v^l. \tag{25}$$

## A.4 DISCUSSION OF ASSUMPTION 1

In Section 3.2, we introduce Assumption 1 to ensure full-rank covariance matrices. A rank-deficient covariance matrix is problematic for differential privacy (DP) as it suggests a complete loss of randomness along certain directions in high-dimensional space. In this section, we discuss when this assumption is likely to hold and potential remedies if it does not.

The covariance matrix of the batch gradient estimator can be expressed as a weighted sum of the transposed Jacobian matrices of output logits with respect to the parameters multiplied by itself.

$$\Sigma_B := \mathrm{Var}(\sum_{d \in B}\hat{g}(d)) = \sum_{d \in B}\mathrm{Var}(\hat{g}(d)) \approx \sum_{d \in B}\frac{\mathcal{L}_0^2}{\sigma^2}(D_\theta v)^\top \cdot D_\theta v \tag{26}$$

Using the Rayleigh quotient, we can show that the minimum eigenvalues of two semi-definite matrices added together must be greater than the minimum eigenvalues of any of them. This indicates that a larger batch size will facilitate the full rank or further increase the minimum eigenvalue. For the same reason, the rejection mechanism is designed.

Consider the case of a single input (batch size of 1) passing through a linear layer with input size $(C, H_{\mathrm{in}})$ and output size $(C, H_{\mathrm{out}})$, where $H_{\mathrm{in}}$ and $H_{\mathrm{out}}$ are the numbers of input and output features, respectively, and $C$ is the number of channels. Denote the input as $x = [x_{i,j}]$ and the output as $v = [v_{i,j}]$. We focus on the weight parameter, $w = [w_{i,j}] \in \mathbb{R}^{H_{\mathrm{out}} \times H_{\mathrm{in}}}$, because the bias part is always full-rank. Flatten the weight and output as $\bar{w} = (\bar{w}_1, ..., \bar{w}_{H_{\mathrm{out}}H_{\mathrm{in}}})$ and $\bar{v} = (\bar{v}_1, ..., \bar{v}_{CH_{\mathrm{out}}})$, where $\bar{w}_{iH_{\mathrm{in}}+j} = w_{i,j}$ and $\bar{v}_{iH_{\mathrm{out}}+j} = v_{i,j}$. Let the Jacobian matrix of the output with respect to the weight be $D = [D_{i,j}] \in \mathbb{R}^{(CH_{\mathrm{out}}) \times (H_{\mathrm{out}}H_{\mathrm{in}})}$, where $[D_{i,j}] = \frac{\partial \bar{v}_i}{\partial \bar{w}_j}$, i.e.,

$$[D]_{kH_{\mathrm{out}}+l, mH_{\mathrm{in}}+n} = \frac{\partial \bar{v}_k H_{\mathrm{out}} + l}{\partial \bar{w}_m H_{\mathrm{in}} + n} = \frac{\partial v_{k,l}}{\partial w_{m,n}}. \tag{27}$$

$D$ is a sparse matrix, where $\frac{\partial v_{k,l}}{\partial w_{m,n}} = 0$ when $l \neq m$ and $\frac{\partial v_{k,m}}{\partial w_{m,n}} = x_{k,n}$. Consequently, the transposed Jacobian matrix multiplied by itself, $D^\top \cdot D$, is a block diagonal matrix with identical blocks. Denote each block as $B(D^\top D) \in \mathbb{R}^{H_{\text{in}} \times H_{\text{in}}}$. Without loss of generality, consider one single block. We have $B(D^\top \cdot D) = x^\top \cdot x$, which has at most $C$ ranks. Ideally, when batch diversity is high, the assumption holds if the batch size exceeds the ratio of input features to input channels. Complex layers like convolutional layers are less prone to rank deficiency due to parameter reuse (e.g., kernel sliding on feature maps). Models like ResNet, where linear layers are a minor component, further mitigate this issue.

In practice, the assumption sometimes fails due to infertile diversity in data or intended small batch size. If so, alternative solutions exist. One option is to alter the location where noise is added. Concretely, we could consider a virtual linear with the input of an identical matrix and the weight of the model parameters. Then, adding noise to the logit of this virtual linear layer equals adding noise to the model parameters directly, and the Jacobian matrix would be the identity matrix, ensuring the full rank. Another approach is to add extra noise directly to the estimated gradient, compensating for randomness deficiencies along its eigenvector directions. This involves calculating the batch's gradient covariance matrix by Equation (27). Next, perform eigendecomposition: $\Sigma_B = Q \cdot \Lambda \cdot Q^{-1}$ and compute the required covariance matrix of the extra noise by $\Sigma_{\text{extra}} = \sigma_0^2 C^2 \mathbb{I} - \text{diag}(\Lambda)/K$, where $\sigma_0$ is the target std scale, $C$ is the clip threshold, and $K$ is the repeat time. After we generate the extra noise with covariance matrix $\Sigma_{\text{extra}}$, we use $Q$ to transform it and then add transformed noise to the estimated gradient of the batch.

## A.5 COMPARISON TO EXISTING DP ZEROTH-ORDER METHODS

Several recent works Liu et al. (2024b); Zhang et al. (2024); Tang et al. (2024) propose DP zeroth-order methods that privatize loss values or estimated gradients obtained via two forward passes in zeroth-order optimization for achieving DP guarantee. Our proposed DP-ULR departs from these methods in the following key aspects:

**Motivation.** Existing approaches achieve differential privacy by introducing additional noise to zeroth-order gradients or losses. In contrast, our work first noticed that forward learning's inherent randomness has the potential for a "free lunch" to provide privacy guarantees. Motivated by this, we propose DP-ULR, which leverages the noise added for gradient estimation in forward learning algorithms to provide privacy guarantees.

**Core Algorithm and Application Scope.** Existing works utilize the traditional zeroth-order method, Simultaneous Perturbation Stochastic Approximation (SPSA), which adds noise to parameters with dimensions significantly higher than logits—often exceeding 100 times. This leads to substantial increases in computational costs (and estimation variance) as the model size grows, limiting their scalability to complex deep-learning models, particularly for training from scratch. Those existing methods are designed for fine-tuning pre-trained models. In contrast, DP-ULR operates directly on logits, enabling the training of deep learning models from scratch and reducing the computational overhead.

**Privacy Bound.** DP-ULR offers superior privacy guarantees compared to methods like ZeroDP Liu et al. (2024b). ZeroDP has the most similar zeroth-order optimization setting to us, involving stochastic gradient descent and repeated sampling. The privacy cost of ZeroDP scales quadratically with the number of repetitions $P$ (Theorem 4.1 in Liu et al. (2024b)), resulting in rapidly increasing privacy costs for large $P$. In contrast, DP-ULR's privacy cost is independent of the number of repetitions, ensuring more robust and scalable privacy protection.

# B DIFFERENTIAL PRIVACY OF DP-ULR

The following theorem is a general form for Theorem 1 and Theorem 2. In SRGM, isotropic Gaussian noise is added to the deterministic output. Then, the variance of output is irrelevant to the size of the sampled batch, and the minimum eigenvalue is the same as the predefined variance $\sigma^2$. In DP-ULR, we ensure $\min \boldsymbol{\lambda}(\sum_{i \in J} \Sigma_{d_i}) \geq \sigma^2$, $\forall J \in 2^{[N]}$ and $|J| \geq N_B$ by our differentially private controller.

**Theorem 4.** *Suppose that $f : \mathbb{D} \to \mathbb{R}^d$ is a randomized function and $f(\cdot)$ follows multivariate Gaussian distribution $\mathcal{N}(\nu_d, \Sigma_d)$ with $\|\nu_d\|_2 \leq 1$, $\forall d \in D$. For $D \in 2^{\mathbb{D}}$ with $|D| \geq \bar{N}$, consider a*

*randomized mechanism $\mathcal{M}$ defined by $\mathcal{M}(D) := \sum_{i \in J} f(d_i)$, where $J \subset [N]$ is a random sample from $[N]$, where $N = |D|$. In the sampling, each $i \in [N]$ is chosen independently with probability $q$, but if the size of $J$ is smaller than $N_B$, it is resampled. Let $\boldsymbol{\lambda}(\cdot)$ denote the spectrum of the matrix. Assume there exist $\sigma \geq 1$ such that $\min \boldsymbol{\lambda}(\sum_{i \in J} \Sigma_{d_i}) \geq \sigma^2$, $\forall J \in 2^{[N]}$ and $|J| \geq N_B$. Then, if $1 \leq N_B \leq q\bar{N}$, $q \leq \frac{1}{5}$, $\sigma \geq 4$, and $\alpha$ satisfy $1 < \alpha \leq \frac{1}{2}\sigma^2 A - 2\ln\sigma$ and $\alpha \leq \frac{\frac{1}{2}\sigma^2 A^2 - \ln 5 - 2\ln\sigma}{A + \ln(q\alpha) + 1/(2\sigma^2)}$, where $A := \ln\left(1 + \frac{1}{q(\alpha-1)}\right)$, the mechanism $\mathcal{M}$ satisfies $(\alpha, \gamma)$-RDP for*

$$\gamma = \frac{q\boldsymbol{p}(N_B - 1; \bar{N}, q)}{1 - \boldsymbol{P}(N_B - 1; \bar{N}, q)} + \frac{2q^2}{\sigma^2}\alpha, \tag{28}$$

*where $\boldsymbol{p}(\cdot, \bar{N}, q)$ and $\boldsymbol{P}(\cdot, \bar{N}, q)$ are defined as the probability mass function and cumulative distribution function of the binomial distribution with parameters $\bar{N}$ and $q$, respectively.*

*Proof.* Consider two adjacent datasets $\mathcal{D} = \{d_i\}_1^N$ and $\mathcal{D}' = \{d_i\}_1^{N+1}$. We want to show that

$$\mathbb{E}_{\omega \sim f_0}[(\frac{f_0(\omega)}{f_1(\omega)})^\lambda] \leq \gamma, \tag{29}$$

$$\text{and } \mathbb{E}_{\omega \sim f_1}[(\frac{f_1(\omega)}{f_0(\omega)})^\lambda] \leq \gamma, \tag{30}$$

for some explicit $\gamma$ to be determined later, where $f_0$ and $f_1$ denote the probability density function of $\mathcal{M}(\mathcal{D})$ and $\mathcal{M}(\mathcal{D}')$, respectively. Here we focus on the former one $\mathbb{E}_{\omega \sim f_0}[(\frac{f_0(\omega)}{f_1(\omega)})^\lambda]$. The other one is similar. By the design of mechanism $\mathcal{M}$, we have

$$f_0(\omega) = c_0 \sum_{J \in 2^{[N]}, |J| \geq L} q^{|J|}(1-q)^{N-|J|}\mu(\omega; \sum_{i \in J}\nu_{d_i}, \sum_{i \in J}\Sigma_{d_i}), \tag{31}$$

where $c_0$ is the normalizing constant and $\mu(\cdot; \nu, \Sigma)$ represents the probability density function of Gaussian distribution with mean $\nu$ and covariance matrix $\Sigma$. To simplify the expression, let us denote $\mu_J(\omega) := \mu(\omega; \sum_{i \in J}\nu_{d_i}, \sum_{i \in J}\Sigma_{d_i})$ for any integer set $J$. Similarly, we have

$$f_1(\omega) = c_1 \sum_{J \in 2^{[N+1]}, |J| \geq L} q^{|J|}(1-q)^{N+1-|J|}\mu_J(\omega)$$

$$= c_1 (( \sum_{J \in 2^{[N]}, |J| = L-1} q^L(1-q)^{N+1-L}\mu_{J \cup \{N+1\}}(\omega)$$

$$+ \sum_{J \in 2^{[N]}, |J| \geq L} q^{|J|}(1-q)^{N-|J|}((1-q)\mu_J(\omega) + q\mu_{J \cup \{N+1\}}(\omega))$$

$$< c_1 \sum_{J \in 2^{[N]}, |J| \geq L} q^{|J|}(1-q)^{N-|J|}\left((1-q)\mu_J(\omega) + q\mu_{J \cup \{N+1\}}(\omega)\right)$$

$$:= \bar{f}_1(w)$$

where $c_1$ is the normalizing constant. Then, we have

$$\mathbb{E}_{\omega \sim f_0}[(\frac{f_0(\omega)}{f_1(\omega)})^\lambda] < \mathbb{E}_{\omega \sim f_0}[(\frac{f_0(\omega)}{\bar{f}_1(\omega)})^\lambda] \leq (\frac{c_0}{c_1})^\lambda \mathbb{E}_{\omega \sim f_0}[(\frac{f_0(\omega)}{((1-q) + q\Gamma_{\nu_{d_{N+1}}})f_0(\omega)})^\lambda], \tag{32}$$

where $\Gamma$ is a translation operator defined as $\Gamma_\epsilon f(\omega) = f(\omega + \epsilon)$. Without loss of generality, $\|\nu_{d_{N+1}}\|_2 = 1$ and $\nu_{d_i} = 0$, $i \neq N+1$. Then, we have

$$\mathbb{E}_{\omega \sim f_0}[(\frac{f_0(\omega)}{f_1(\omega)})^\lambda]$$

$$\leq (\frac{c_0}{c_1})^\lambda \mathbb{E}_{\omega \sim f_0}[(\frac{\sum_{J \in 2^{[N]}, |J| \geq L} q^{|J|}(1-q)^{N-|J|}\mu(\omega; 0, \sum_{i \in J}\Sigma_{d_i})}{((1-q) + q\Gamma_{\nu_{d_{N+1}}})\sum_{J \in 2^{[N]}, |J| \geq L} q^{|J|}(1-q)^{N-|J|}\mu(\omega; 0, \sum_{i \in J}\Sigma_{d_i})})^\lambda]$$

$$\leq (\frac{c_0}{c_1})^\lambda \mathbb{E}_{\omega \sim f_0}[(\frac{\sum_{J \in 2^{[N]}, |J| \geq L} q^{|J|}(1-q)^{N-|J|}\mu(\omega; 0, \sigma^2\mathbb{I})}{((1-q) + q\Gamma_{\nu_{d_{N+1}}})\sum_{J \in 2^{[N]}, |J| \geq L} q^{|J|}(1-q)^{N-|J|}\mu(\omega; 0, \sigma^2\mathbb{I})})^\lambda]$$

$$= (\frac{c_0}{c_1})^\lambda \mathbb{E}_{\omega \sim f_0}[(\frac{\mu(\omega; 0, \sigma^2\mathbb{I})}{((1-q) + q\Gamma_{\nu_{d_{N+1}}})\mu(\omega; 0, \sigma^2\mathbb{I})})^\lambda].$$

Without loss of generality, $\nu_{d_{N+1}} = e_1$. Then, in the above equation, the numerator distribution $\mu(\omega; 0, \sigma^2 \mathbb{I})$ and denominator distribution $((1-q) + q\Gamma_{\nu_{d_{N+1}}})\mu(\omega; 0, \sigma^2 \mathbb{I})$ are identical except for the first coordinate and hence we have a one-dimensional problem. Specifically, we have

$$\mathbb{E}_{\omega \sim f_0}[(\frac{f_0(\omega)}{f_1(\omega)})^\lambda] \leq (\frac{c_0}{c_1})^\lambda \mathbb{E}_{\omega \sim \mu_0}[(\frac{\mu_0}{((1-q) + q\Gamma_1)\mu_0})^\lambda], \tag{33}$$

where $\mu_0$ denotes the probability density function of $\mathcal{N}(0, \sigma^2)$. Notice that

$$\mathbb{E}_{\omega \sim f_0}[(\frac{f_0(\omega)}{f_1(\omega)})^\lambda] = \mathbb{E}_{\omega \sim f_1}[(\frac{f_0(\omega)}{f_1(\omega)})^{\lambda+1}]. \tag{34}$$

Then, we have

$$D_\alpha(\mathcal{M}(D) \parallel \mathcal{M}(D')) = \frac{1}{\alpha - 1} \ln \mathbb{E}_{\omega \sim f_1} \left( \frac{f_0(\omega)}{f_1(\omega)} \right)^\alpha \tag{35}$$

$$\leq \frac{1}{\alpha - 1} \ln \left[ (\frac{c_0}{c_1})^{\alpha-1} \mathbb{E}_{\omega \sim ((1-q)+q\Gamma_1)\mu_0}(\frac{\mu_0}{((1-q) + q\Gamma_1)\mu_0})^\alpha \right] \tag{36}$$

$$= \ln \frac{c_0}{c_1} + D_\alpha(\mu_0 \parallel ((1-q) + q\Gamma_1)\mu_0) \tag{37}$$

Using the existing result from Mironov et al. (2019), we can derive

$$D_\alpha(\mathcal{M}(D) \parallel \mathcal{M}(D')) \leq \ln \frac{c_0}{c_1} + \frac{2q^2}{\sigma^2}\alpha, \tag{38}$$

when $q \leq \frac{1}{5}$, $\sigma \geq 4$, and $\alpha$ satisfy $1 < \alpha \leq \frac{1}{2}\sigma^2 A - 2\ln\sigma$ and $\alpha \leq \frac{\frac{1}{2}\sigma^2 A^2 - \ln 5 - 2\ln\sigma}{A + \ln(q\alpha) + 1/(2\sigma^2)}$, where $A := \ln\left(1 + \frac{1}{q(\alpha-1)}\right)$. Particularly, we have

$$\frac{c_0}{c_1} = 1 + c_0 \binom{N}{L-1} q^L (1-q)^{N+1-L} = 1 + \frac{qp(N_B - 1; N, q)}{1 - P(N_B - 1; N, q)}. \tag{39}$$

If $N_B \leq q\bar{N}$, we have

$$\frac{c_0}{c_1} \leq 1 + \frac{qp(N_B - 1; \bar{N}, q)}{1 - P(N_B - 1; \bar{N}, q)} \tag{40}$$

Finally, we have, if $1 \leq N_B \leq q\bar{N}$, $q \leq \frac{1}{5}$, $\sigma \geq 4$, and $\alpha$ satisfy $1 < \alpha \leq \frac{1}{2}\sigma^2 A - 2\ln\sigma$ and $\alpha \leq \frac{\frac{1}{2}\sigma^2 A^2 - \ln 5 - 2\ln\sigma}{A + \ln(q\alpha) + 1/(2\sigma^2)}$, where $A := \ln\left(1 + \frac{1}{q(\alpha-1)}\right)$,

$$D_\alpha(\mathcal{M}(D) \parallel \mathcal{M}(D')) \leq \frac{qp(N_B - 1; \bar{N}, q)}{1 - P(N_B - 1; \bar{N}, q)} + \frac{2q^2}{\sigma^2}\alpha. \tag{41}$$

$\square$

Then, directly using the composition theorem of RDP, we obtain that with certain conditions on parameters, the RDP bound of our DP-ULR is

$$\gamma = \frac{Tq\boldsymbol{p}(N_B - 1; \bar{N}, q)}{1 - \boldsymbol{P}(N_B - 1; \bar{N}, q)} + \frac{2Tq^2}{\sigma_0^2}\alpha. \tag{42}$$

## C  MORE EXPERIMENTS

### C.1  DIFFERENT MODEL SIZES

We conduct ablation experiments to analyze the relationship between noise-redundancy impairment and model size by evaluating three configurations of MLP models: small, medium, and large. The parameter count for MLP (medium) and MLP (large) is approximately 2.5 and 5 times that of MLP (small), respectively. We test both DP-ULR and DP-SGD under a high noise scale (target std level) of $\sigma_0 = 8$ with batch sizes $B = 64, 128, 256$. All other hyperparameters remain consistent with those

| Batch size | $\sigma_0$ | Method Model | DP-ULR Training acc.(%) | Valid acc. (%) | DP-SGD Training acc.(%) | Valid acc. (%) |
|---|---|---|---|---|---|---|
| 64 | 8 | MLP(small) | $57.88_{\pm 4.53}$ | $58.65_{\pm 6.32}$ | $33.63_{\pm 0.65}$ | $32.07_{\pm 4.17}$ |
| | | MLP(medium) | $58.58_{\pm 4.69}$ | $59.53_{\pm 4.38}$ | $42.98_{\pm 5.83}$ | $43.81_{\pm 6.30}$ |
| | | MLP(large) | $40.10_{\pm 5.87}$ | $41.42_{\pm 7.03}$ | $28.75_{\pm 3.90}$ | $24.78_{\pm 6.28}$ |
| 128 | 8 | MLP(small) | $73.04_{\pm 1.37}$ | $74.24_{\pm 1.19}$ | $69.49_{\pm 0.71}$ | $70.53_{\pm 0.57}$ |
| | | MLP(medium) | $67.25_{\pm 0.91}$ | $69.00_{\pm 1.39}$ | $66.22_{\pm 1.61}$ | $67.98_{\pm 1.81}$ |
| | | MLP(large) | $66.48_{\pm 2.13}$ | $67.75_{\pm 3.00}$ | $65.87_{\pm 0.81}$ | $66.15_{\pm 1.49}$ |
| 256 | 8 | MLP(small) | $79.27_{\pm 0.93}$ | $80.94_{\pm 1.12}$ | $85.11_{\pm 0.34}$ | $86.03_{\pm 0.46}$ |
| | | MLP(medium) | $76.45_{\pm 0.42}$ | $78.48_{\pm 0.43}$ | $83.93_{\pm 0.52}$ | $84.96_{\pm 0.68}$ |
| | | MLP(large) | $76.25_{\pm 0.34}$ | $78.28_{\pm 0.80}$ | $84.29_{\pm 0.52}$ | $85.13_{\pm 0.89}$ |

Table C1: The classification accuracy of MLP of different sizes on the MNIST dataset.

specified in the Section 4.2. Each experiment is repeated five times with different random seeds, and the mean and standard deviations are reported in Table C1.

The results indicate that with smaller batch sizes, the performance advantage of DP-ULR over DP-SGD diminishes as model size increases. This trend may be attributed to noise redundancy, stemming from two factors: (1) DP-ULR's privacy cost is influenced by the smallest singular value of the Jacobian matrix, and (2) the non-isotropic variance of our gradient proxy, which tends to grow with model size. However, this phenomenon is mitigated as batch size increases. For batch sizes of 128 and 256, the performance gap between DP-ULR and DP-SGD remains consistent regardless of model size. This stabilization is likely due to the increased sample diversity with larger batches, which reduces the non-isotropy of the

