# OpenReview forum: "Forward Learning with Differential Privacy"
_ICLR.cc/2025/Conference — Submitted to ICLR 2025_

### Official Review · Reviewer_dZ7t · 2024-10-27

**Soundness:** 2
**Presentation:** 2
**Contribution:** 2
**Rating:** 3
**Confidence:** 4

**Summary:**

The paper introduces a new algorithm, the Differential Private Unified Likelihood Ratio method (DP-ULR), which enhances privacy protection in deep learning by utilizing forward-learning techniques instead of traditional backpropagation. DP-ULR integrates noise during the forward pass. Theoretical analysis and experiments claim to show that DP-ULR achieves better performance than DP-SGD.

**Strengths:**

1. This paper proposes a novel approach to DP training other than the popular DP-SGD
2. The paper conducts reasonable theoretical analysis and experiments.

**Weaknesses:**

I think the proposed approach is interesting but I have some questions about it.

1. Misleading DP-SGD baseline: It is common sense that the larger the batch size is, the better performance we can achieve with DP-SGD. See Dormann et al. (https://arxiv.org/pdf/2110.06255), Figure 2 of De et al. (https://arxiv.org/pdf/2204.13650) and Figure 4(b) of Yu et al. (https://arxiv.org/pdf/2306.08842). Why do the results in Table 1 of this paper give the opposite conclusion? Let's put your proposed method aside, and just look at the results of DP-SGD, could authors explain why the performance drops for large batch size when the noise level is the same? Could it be the learning rate set improperly?

2. The results presented in Table 1 reveal that the performance gap between DP-ULR and DP-SGD varies and is relatively small in some instances. To draw more reliable conclusions, it would be beneficial to report the average accuracy across multiple runs, along with the standard deviation. Additionally, the performance of MLP on MNIST appears notably low, even at reduced noise levels.

3. For the entire experiment, it makes more sense to utilize more popular models such as ResNet-18 or Wide-ResNet-16-4 or 40-4 which is widely used in literature such as De et al. Also, it is very important to show whether the proposed method can be applied to the complex model which is more complex than ResNet-5 used in the paper.  BTW, it is not very clear from Figure 4 what is the final test accuracy for DP-SGD and DP-ULR. I can see they are close to each other. Does it mean that the proposed method can not achieve better utility than DP-SGD when dealing with a large model?

**Questions:**

Please see the weakness.

---

> ### Author Response · Authors · 2024-11-28
>
> **[W1] Misleading DP-SGD baseline**
>
> **[RW1]:** We understand your concern about the DP-SGD baseline and appreciate the opportunity to clarify. For DP-SGD experiments, we used the open-source OPACUS library with default settings for MNIST. We tested learning rates of 0.1 and 0.01, reporting the best-performing results (0.1 for all cases).
>
> We agree that larger batch sizes generally improve the utility-privacy tradeoff in DP-SGD, as noted in previous literature. Specifically, performance was improved with a larger batch size when fixing privacy cost $(\epsilon, \delta)$. This improvement arises from increasing suitable noise scale $\sigma$ with a larger batch size. In contrast, higher noise scales significantly impair training with small batch sizes. In our experiments, we can also observe this improvement. For example, the privacy cost of training with $B=64$ and $\sigma=2$ is similar to, actually higher than, that of training with $B=500$ and $\sigma=8$. The valid acc. of the former setting is $68.73\%$, while that of the latter setting is $88.16\%$, indicating a better performance on both accuracy and privacy cost.
>
> Besides, we kindly remind the reviewer that in our experiments, the performance of DP-SGD increases or for large batch size in most of case of $\sigma$ (2,4,8), and evidently drops only when $\sigma = 0.5$, where training resembles a non-privatized setting. We attribute this specific drop to hyperparameter or parameter interaction effects rather than inherent limitations of DP-SGD.
>
> To further ensure the fairness of the baseline results, we extended our DP-SGD experiments across a wider range of learning rates (0.1, 0.05, 0.01, 0.005, 0.001) with larger batch sizes ($B=200, 500$). The results, presented below, show that the learning rate of 0.1, which was used in our paper, consistently achieves the best performance across almost all settings:
>
> ||Learning rate|0.001|0.005|0.01|0.05|0.1|0.001|0.005|0.01|0.05|0.1|
> |-|-|-|-|-|-|-|-|-|-|-|-|
> |Batch size|$\sigma_0$|Training acc.(\%)|Training acc.(\%)|Training acc.(\%)|Training acc.(\%)|Training acc.(\%)|Valid acc.(\%)|Valid acc.(\%)|Valid acc.(\%)|Valid acc.(\%)|Valid acc.(\%)|
> |200|0.50|41.51|59.68|75.31|88.77|90.47|42.18|59.91|76.04|88.89|90.68|
> |200|1.00|23.28|67.94|76.38|88.44|90.54|23.40|69.25|77.20|88.98|90.77|
> |200|2.00|26.71|64.98|74.64|88.14|90.63|27.01|65.73|75.67|88.41|90.95|
> |200|4.00|13.76|61.31|75.97|88.08|89.07|14.50|61.84|76.76|88.95|89.63|
> |200|8.00|29.39|64.76|75.00|85.29|81.07|30.48|65.70|75.41|86.39|81.92|
> |500|0.50|11.73|44.50|62.52|82.87|87.56|11.49|44.15|64.44|83.54|87.98|
> |500|1.00|9.93|55.94|58.54|83.22|87.56|10.12|56.90|58.99|83.84|87.98|
> |500|2.00|14.34|55.95|62.97|85.20|87.59|14.42|57.21|64.46|85.40|88.00|
> |500|4.00|16.00|50.91|60.89|83.86|87.66|15.87|50.17|61.74|84.24|87.97|
> |500|8.00|16.45|55.40|63.38|84.44|87.68|16.80|56.14|65.09|84.95|88.16|
>
>
> **[W2] Report average and std for multiple runs**
>
> **[RW2]:** Thank you for the suggestion. We have rerun our experiments five times and calculated the average and standard deviation. These results have been incorporated into the Table 1 of the revised manuscript.
>
> **[W3] Experiments with more popular models**
>
> **[RW3]:** We appreciate the suggestion to extend our study to more complex architectures. This work investigates the connection between forward learning and differential privacy. While previous research has focused on improving performance, particularly in terms of gradient estimation accuracy, little attention has been given to fully exploring the potential of forward learning’s inherent characteristics. Our study aims to address this gap from a theoretical perspective, specifically how forward learning can be leveraged to provide privacy to neural networks in a natural way.
>
> In response to your question, applying forward learning to large-scale neural networks remains a significant challenge. While current studies [1,2] suggest that forward learning algorithms can be used to fine-tune large models, to the best of our knowledge, no work has yet extended the application of forward learning to train large networks from scratch and achieve comparable performance. While Deepzero [3] claims success in this area, they rely on pruning techniques, and there are reproducibility issues, as indicated in their repository's issue channel. This suggests that there is still much work to be done in this direction.
>
> We hope that our work can provide further support for forward learning research, with the added confidence that it inherently possesses the beneficial properties of differential privacy. This is another key message we aim to convey through our work.
>
> [1] T. Ren et al., "FLOPS: Forward Learning with OPtimal Sampling." *arXiv preprint, 2024.*
>
> [2] S. Malladi et al., "Fine-Tuning Language Models with Just Forward Passes." *arXiv preprint, 2024.*
>
> [3] A. Chen et al., "DeepZero: Scaling up Zeroth-Order Optimization for Deep Model Training." *arXiv preprint, 2024.*

---

> > ### Comment · Reviewer_dZ7t · 2024-12-01
> >
> > Thanks for the experiments. I still have a few comments.
> >
> > 1. Why not use a large learning rate for DP-SGD for example like 1 or 2? It is shown in many papers that DP-SGD needs a large learning rate to get better performance (see Figure 8 at https://arxiv.org/pdf/2204.13650). I think it is important to do a fair comparison between the proposed method and baseline for example do the correct hyperparameter search even for the baseline.
> > 2. Your proposal seems to suggest that your method might not be practical for large-scale models or even for MLPs with large batch sizes, as DP-SGD appears to be more effective according to Table C1. Could you clarify the specific scenarios or applications where your method is advantageous, aside from its use with toy datasets and models? Additionally, is there potential for your method to be adapted for practical applications, or is it primarily intended for theoretical exploration? I'm not questioning the novelty or interest of your approach; rather, I'm interested in understanding its practical utility beyond theoretical research.
> > 3. BTW appendix C.1 is not finished.

---

> ### Author Response · Authors · 2024-12-02
>
> **[R1]:** We appreciate your insightful comment regarding learning rates. Initially, we experimented with different learning rates and observed that the performance of DP-SGD at larger learning rates could rise but then suffers from more severe degradation, particularly with larger noise scales or smaller batch sizes. In our following hyperparameter search, we found that a learning rate of 0.1 offered the best overall performance across various settings. Therefore, we exhibit the results of the current learning rate in the paper, which, although it may not be optimal for every setting, represents a carefully tuned choice that balances performance across the board.
>
> To address your suggestion, we conducted additional experiments with larger learning rates (1 and 2). The results, shown below, indicate that while these larger rates do not consistently outperform a learning rate of 0.1, which achieves the best average performance in most cases, they do yield improvements in some cases (e.g., larger batch sizes and lower noise levels). Thank you again for your instructive suggestion. We will include these additional results in the revised manuscript to provide a more comprehensive comparison.
>
>
> ||**Learning rate**|0.1|1|2|0.1|1|2|
> |---|---|---|---|---|---|---|---|
> |Batch size|$\sigma_0$|Training acc.(\%)|Training acc.(\%)|Training acc.(\%)|Valid acc.(\%)|Valid acc.(\%)|Valid acc.(\%)|
> |64|0.5|**93.94**|83.98|77.86|**94.14**|84.36|76.20|
> |64|1|**90.90**|70.12|32.94|**91.20**|65.54|11.65|
> |64|2|**83.93**|30.68|12.59|**84.93**|12.60|11.67|
> |64|4|**67.33**|12.59|11.88|**68.73**|12.11|12.07|
> |64|8|**33.63**|12.95|12.85|**32.07**|12.39|12.39|
> |200|0.5|90.47|**94.97**|93.52|90.68|**94.80**|93.40|
> |200|1|**90.54**|89.88|87.05|**90.77**|90.56|86.79|
> |200|2|**90.63**|82.42|73.21|**90.95**|82.74|73.08|
> |200|4|**89.07**|65.00|43.38|**89.63**|66.02|26.14|
> |200|8|**81.07**|31.87|13.87|**81.92**|17.22|11.77|
> |500|0.5|87.56|95.20|**96.67**|87.98|95.19|**96.37**|
> |500|1|87.56|**94.73**|94.66|87.98|**94.92**|94.92|
> |500|2|87.59|**92.29**|89.52|88.00|**92.54**|90.32|
> |500|4|**87.66**|85.83|82.38|**87.97**|86.53|83.48|
> |500|8|**87.68**|74.57|61.29|**88.16**|77.35|64.65|
>
>
> Besides, we kindly remind the reviewer that, in the paper, we also didn't do hyperparameter searching for our DP-ULR in every setting of batch size and noise scale and applied consistent hyperparameters. For further enhanced comparisons, we selected several cases (not all cases due to the time limit), particularly the specific cases where DP-SGD benefits from larger learning rates, to showcase the improvements of DP-ULR with similar hyperparameter tuning. For example, with a slightly larger learning rate of 0.02, DP-ULR achieves the following performance improvements, which are still comparable to DP-SGD:
> - $93.32\%$ training acc. and $93.36\%$ validation acc. for $B=200$, $\sigma_0=0.5$.
> - $94.60\%$ training acc. and $94.63\%$ validation acc. for $B=500$, $\sigma_0=0.5$.
> - $92.65\%$ training acc. and $92.71\%$ validation acc. for $B=500$, $\sigma_0=1$.
> - $91.35\%$ training acc. and $92.04\%$ validation acc. for $B=500$, $\sigma_0=2$.
>
>
> **[R2]:** Thank you for your thoughtful question and interest in our work. DP-ULR primarily aims to enable differential privacy in non-backpropagation (non-BP) algorithms. Consequently, the training capacity of DP-ULR aligns more closely with its foundational method, ULR, rather than BP. Moreover, DP-ULR inherits ULR's advantages, such as suitability for training black-box or non-differentiable models.
>
> One significant practical application of DP-ULR is fine-tuning scenarios where the model backbone remains frozen. While DP-SGD requires gradient computations through frozen components using BP's chain rule, DP-ULR avoids this constraint. This makes DP-ULR particularly valuable when dealing with black-box backbones (e.g., GPT). In such cases, DP-SGD becomes inapplicable, whereas DP-ULR can perform privacy-preserving fine-tuning, such as through LoRA. This is critical for leveraging foundation models like GPT, which carry rich pretraining knowledge and are often fine-tuned with sensitive, domain-specific user data. By enabling private fine-tuning, DP-ULR directly supports real-world deployment needs.
>
> Although this paper focuses on establishing the theoretical and practical foundations of DP-ULR, we believe it opens the door to scaling forward-learning-based approaches for practical applications, such as secure fine-tuning large-scale foundation models. We will clarify these potential applications in the revised manuscript.
>
>
>
> **[R3]:** Thank you for your kind reminder. It was a compiling issue of tex. We will revise this with several missing words. The complete sentence is, "This stabilization is likely due to the increased sample diversity with larger batches, which reduces the non-isotropy of the **variance and minimizes its impact on training performance.**"

---

### Official Review · Reviewer_pQjz · 2024-11-01

**Soundness:** 3
**Presentation:** 4
**Contribution:** 3
**Rating:** 6
**Confidence:** 4

**Summary:**

In this paper, the authors theoretically analyze the feasibility of DP with noise introduced by forward learning and propose a ULR variant, DP-ULR. The paper presents the novel batch sampling operation with rejection, which is used in DP-ULR. The paper also analyzes the performance and privacy guarantee of DP-ULR in both theoretical and experimental terms.

**Strengths:**

- The authors discovered that forward learning's inherent randomness can be utilized to achieve DP, which is an enlightening combination of noise introduced by both DP and forward learning.

- The paper introduces a novel sampling-with-rejection technique for DP-ULR, which provides a quadratic privacy amplification. Realizing such a technique may introduce impairment on the privacy cost, the authors propose an assumption for DP-ULR and design experiments to demonstrate that such an impairment is minimal.

- The paper is well written, clearly presenting the motivation for combining DP and forward learning. Besides, the theoretical analysis part is presented in conjunction with traditional methods, which makes it easy for readers to follow.

**Weaknesses:**

- The authors mention that DP-ULR inherits certain advantages over backpropagation-based DP-SGD, such as suitability for non-differentiable or black-box settings, high parallelizability, and efficient pipeline design. However, the paper lacks experiments on non-differentiable or black-box settings to support these ideas. Even though current hardware or software implementation may limit the efficiency of DP-ULR, the authors do not provide an asymptotic complexity analysis of DP-ULR either.

**Questions:**

- How does DP-ULR perform on non-differentiable modules or in black-box settings as mentioned in the abstract?
- Could the authors provide an asymptotic complexity analysis of DP-ULR?
- The authors mention that non-isotropy may introduce redundant noise to DP-ULR, which may impair the accuracy of the gradient estimation. The reviewer wonders about the proportional relationship between this impairment and the number of model parameters. For example, experiments comparing DP-ULR and DP-SGD on different model sizes could be added to the paper to illustrate this relationship. It would help to clarify whether DP-ULR's performance degrades relative to DP-SGD as models become larger.

---

> ### Author Response · Authors · 2024-11-28
>
> **[W1 & Q1] Lack of experiments and analysis on non-differentiable or black-box settings**
>
> **[RW1 & RQ1]:** Thank you for your thoughtful feedback. In the abstract, we initially highlight the general advantages of ULR to show the rationale for using it as the basis for our study. However, these advantages are not intended to be the primary selling points of DP-ULR, which focuses on leveraging the inherent randomness of forward learning algorithms to achieve differential privacy. Our work aims to inspire further exploration into the application of DP forward learning, including non-differentiable and black-box settings.
>
> Nonetheless, our experiments partly demonstrate DP-ULR's capability in black-box settings. Specifically, while all layers are trained simultaneously, the gradient estimations of different layers are independent. When noise is added to the output of a layer for gradient estimation, other layers effectively function as black boxes. In contrast, backpropagation relies on computation graphs and cannot obtain gradients without access to subsequent layers. For non-differentiable modules, such as spiking neural networks (SNNs), ULR has been shown to outperform SGD, which requires smoothing approximations for backpropagation.
>
> In the initial submission, we acknowledged the absence of direct experiments in these settings in the limitations section. In the revised manuscript, we have further revised the abstract and introduction to clarify that DP-ULR's suitability for non-differentiable or black-box settings is a potential rather than a demonstrated advantage. Thank you again for your constructive suggestions.
>
> **[W1 & Q2] Complexity analysis of DP-ULR**
>
> **[RW1 & RQ2]:** Thank you for the suggestion. For an MLP with $L$ layers and $N$ neurons per layer, the complexity of DP-ULR varies based on parallelism:
>
> - **No Interlayer Parallelism:** The forward stage requires $L$ independent forward passes, resulting in a complexity of $O(L^2N^2)$. The gradient computation (backward stage) has a complexity of $O(LN^2)$.
> - **Full Interlayer Parallelism:** A favorable property inherited from ULR is that both forward passes and gradient computations can be parallelized, reducing forward-stage complexity to $O(LN^2)$ and gradient computation complexity to $O(N^2)$.
>
> Due to the same request, please refer to the response to Reviewer ZJUY for a detailed explanation.
>
> **[Q3] Relationship between noise redundancy and model size**
>
> **[RQ3]:** Thank you for the constructive suggestion. We conducted additional experiments on three MLP configurations: small, medium (2.5x parameters), and large (5x parameters). Both DP-ULR and DP-SGD were tested with a noise scale $\sigma_0 = 8$ and batch sizes $B = 64, 128, 256$. Each experiment was repeated five times, and the mean and standard deviations are reported below:
>
> ||||DP-ULR|DP-ULR|DP-SGD|DP-SGD|
> |------------|------------|-------------|-------------------|-------------------|-------------------|-------------------|
> | Batch size | $\sigma_0$ | Model | Training acc.(\%) | Valid acc. (\%) | Training acc.(\%) | Valid acc. (\%) |
> | 64 | 8 | MLP(small)  | 57.88$_{\pm4.53}$ | 58.65$_{\pm6.32}$ | 33.63$_{\pm0.65}$ | 32.07$_{\pm4.17}$ |
> | 64 | 8 | MLP(medium) | 58.58$_{\pm4.69}$ | 59.53$_{\pm4.38}$ | 42.98$_{\pm5.83}$ | 43.81$_{\pm6.30}$ |
> | 64 | 8 | MLP(large)  | 40.10$_{\pm5.87}$ | 41.42$_{\pm7.03}$ | 28.75$_{\pm3.90}$ | 24.78$_{\pm6.28}$ |
> | 128 | 8 | MLP(small)  | 73.04$_{\pm1.37}$ | 74.24$_{\pm1.19}$ | 69.49$_{\pm0.71}$ | 70.53$_{\pm0.57}$ |
> | 128 | 8 | MLP(medium) | 67.25$_{\pm0.91}$ | 69.00$_{\pm1.39}$ | 66.22$_{\pm1.61}$ | 67.98$_{\pm1.81}$ |
> | 128 | 8 | MLP(large)  | 66.48$_{\pm2.13}$ | 67.75$_{\pm3.00}$ | 65.87$_{\pm0.81}$ | 66.15$_{\pm1.49}$ |
> | 256 | 8 | MLP(small)  | 79.27$_{\pm0.93}$ | 80.94$_{\pm1.12}$ | 85.11$_{\pm0.34}$ | 86.03$_{\pm0.46}$ |
> | 256 | 8 | MLP(medium) | 76.45$_{\pm0.42}$ | 78.48$_{\pm0.43}$ | 83.93$_{\pm0.52}$ | 84.96$_{\pm0.68}$ |
> | 256 | 8 | MLP(large)  | 76.25$_{\pm0.34}$ | 78.28$_{\pm0.80}$ | 84.29$_{\pm0.52}$ | 85.13$_{\pm0.89}$ |
>
> When the batch size is small, DP-ULR's advantage over DP-SGD decreases as the model size increases. This is likely due to DP-ULR's reliance on the minimum spectrum of the Jacobian matrix, introducing noise redundancy from non-isotropic variance, which scales with model size, as hypothesized by the reviewer. However, with larger batch sizes (128 and 256), the performance gap between DP-ULR and DP-SGD stabilizes, likely because increased sample diversity reduces the extent of non-isotropy, mitigating its impact. Besides, we can see that both DP-ULR and DP-SGD show uncommonly decreased performance with increased model size. We attribute this degradation to untuned training hyperparameters related to model size and argue that this doesn‘t affect our previous analysis.
>
> We have included these results and discussions in Appendix C.5 of the revised manuscript to indicate the constructive insight pointed out by the reviewer.

---

> ### Author Response · Authors · 2024-12-02
>
> Dear Reviewer pQjz,
>
> Thank you for your time and thoughtful review. With one day remaining in the discussion phase, we kindly request your feedback on our responses and look forward to your updated evaluation. If you have any additional questions or concerns, please feel free to let us know.

---

> > ### Comment · Reviewer_pQjz · 2024-12-03
> >
> > This reviewer would like to thank the author(s) for their detailed response.
> > 1. The author's responses to Q1 and Q2, as well as the revisions made to the manuscript, have addressed some of the shortcomings in the original text.
> > 2. However, the additional experimental results indicate that DP-ULR performs poorly with larger models and larger batch sizes (which the author attributes to untuned training hyperparameters). Given that existing studies [1] have highlighted that DP-SGD achieves higher accuracy with larger batch sizes (e.g., 4096), the reviewer believes that this performance degradation of DP-ULR could impact its applicability.
> >
> > [1] De, S., Berrada, L., Hayes, J., Smith, S. L., & Balle, B. (2022). Unlocking high-accuracy differentially private image classification through scale. arXiv preprint arXiv:2204.13650.

---

> > > ### Author Response · Authors · 2024-12-03
> > >
> > > Thank you for your additional comments and for engaging with our additional experimental results. We appreciate the opportunity to clarify the applicability and potential use cases of DP-ULR, particularly in comparison to DP-SGD.
> > >
> > > We would like to emphasize the expectations regarding the role of DP-ULR kindly. Our method is not intended to replace DP-SGD in conventional training pipelines but rather to enable differential privacy in settings where DP-SGD is inapplicable or impractical. These include scenarios requiring compatibility with black-box models or specialized fine-tuning tasks on sensitive, domain-specific data. While DP-ULR's training process can be more challenging without leveraging model structure (a "cost to pay"), it uniquely enables flexibility for scenarios that demand capabilities beyond what DP-SGD can offer.
> > >
> > > Besides, we kindly remind the reviewer that our experiments with varying batch sizes (Table. 1 in the paper) do not indicate a decline in performance as batch size increases. Instead, larger batch sizes consistently lead to improved performance under both the same noise scale and the same privacy budget. Furthermore, as demonstrated in our latest additional experiments (in the last paragraph of [R1] of the latest response to Reviewer dZ7t), the performance of DP-ULR with large batch sizes can be significantly enhanced through slight hyperparameter tuning, further showing its potential in such scenarios. Regarding experiments with different model sizes, we observed that DP-SGD also exhibits an unusual performance decline as model size increases. This suggests that the observed degradation of both two methods may come from suboptimal parameter settings (model architecture or training hyperparameters) rather than an inherent limitation. Due to the time limit, we were unable to conduct additional experiments during the rebuttal phase to investigate this issue thoroughly. Therefore, we exhibited the results in Table C.1 as they currently stand.
> > >
> > > We will revise the manuscript to better emphasize these trade-offs and clarify that DP-ULR is a complementary approach to DP-SGD and is tailored for distinct applications. Thank you again for your valuable feedback.

---

### Official Review · Reviewer_RAbX · 2024-11-02

**Soundness:** 2
**Presentation:** 3
**Contribution:** 2
**Rating:** 5
**Confidence:** 3

**Summary:**

Forward-learning algorithm is a kind of non-DP algorithm that updates the model parameters without back-propagation, which has advantage on parallelizability and black-box settings.  The paper introduces DP-ULR which attempts to ensure differential privacy (DP) in forward-learning by utilizing randomness in the forward pass. The method leverages a unique batch sampling approach with rejection and integrates a privacy controller to dynamically adjust noise for privacy-cost management. Their experimental results indicate the proposed method shares a comparable performance as the DPSGD algortihm.

**Strengths:**

- The research problem is clearly motivated by the limitation of DP-SGD, such as its infeasibility of black-box modules and differentiable requirement.
- The paper is well-written, with clear demonstration and algorithm details.
- The results are comparable to DPSGD, but without the need to perform back-propagation.

**Weaknesses:**

- **Insufficient differential privacy protection**: This work aims to provide a final $(\epsilon, \delta)$-DP, which by definition should bound the influence of a sample in the dataset, including both feature \(x\) and label \(y\). However, in Line 199 of Algorithm 1, for each sample $(x_i, y_i) \in B_t$ in a batch, only the output derived from the input $x_i$ is perturbed while $y_i$ is used with its ground-truth value. Thus, it might be unconvincing that the proposed algorithm guarantees DP.

- **Data-dependent noise magnitude**: As equation (8) shows, the noise magnitude is determined by calculating the minimum spectrum of a matrix, which is non-privatized. Therefore, calculating the sigma itself violates privacy, and the overall differential privacy guarantee might be problematic.

- **The efficiency advantage is unclear compared to ZeroDP**: Starting from Line 278, the algorithm requires one more forward pass to compute the Jacobian matrix and loss without any perturbation. Thus, in total, there are two forwards for every sample. There are several works that privatize loss values obtained in two forward passes in zeroth-order optimization for achieving DP guarantee on protecting every sample. If this work also requires two forward passes, what is the advantage compared to the line of ZeroDP works? In Section 3.4, the authors discuss the efficiency advantage compared to DP-SGD. However, there lacks discussion to compare the advantage to other forward-based DP algorithms, such as the following references:
  - Differentially Private Zeroth-Order Methods for Scalable Large Language Model Finetuning
  - Faster Differentially Private Convex Optimization via Second-Order Method
  - Private Fine-tuning of Large Language Models with Zeroth-Order Optimization

**Questions:**

- How does this compare with the line of work on ZeroDP?
- Why does using the ground truth $y_i$ still yield a sample-level DP guarantee?
- Does using the non-privatized minimum spectrum of a matrix to estimate $\sigma$ violate privacy?

---

> ### Author Response · Authors · 2024-11-28
>
> **[W1 & Q2] Insufficient differential privacy protection**
>
> **[RW1 & RQ2]:** Thank you for your insightful feedback. We agree that perturbing both inputs and labels could enhance the differential privacy guarantee, and this is a valuable direction for future exploration. However, we argue that perturbing the input $x$ alone is sufficient to guarantee DP. In deep learning, differential privacy hinges on the randomness of the algorithm's output (i.e., learned parameters), which are computed iteratively through gradients derived from both inputs and labels. By perturbing the input, we effectively randomize the gradients, thus ensuring privacy. Our theoretical analysis focuses on the distribution of output gradients when only the input is perturbed, providing solid DP guarantees. We choose to perturb the input rather than the label due to the forward learning algorithm's reliance on input perturbations for gradient estimation and the inherent randomness this introduces. This approach aligns with our goal of translating inherent randomness into differential privacy guarantees.
>
> **[W2 & Q3] Non-privatized computation of the minimum spectrum**
>
> **[RW2]:** We appreciate your concern regarding the non-privatized computation of the minimum spectrum. However, we argue that this operation does not affect the DP guarantee. Instead of influencing the expectation of the output gradient, the purpose of this computation is to ensure a consistent lower bound on variance, which is utilized to derive the privacy guarantee. This is comparable to gradient norm computation for clipping in DP-SGD, which is also non-privatized but does not affect the DP analysis. In our case, the privacy guarantee is derived solely from the distribution of output gradients, independent of the intermediate non-privatized computations.
>
>
> **[W3 & Q1] (Efficiency) comparison to ZeroDP and similar works**
>
> **[RW3 & RQ1]:** Thank you for highlighting the need for a clear comparison with ZeroDP [1] and similar works [2,3]. While existing DP zeroth-order methods, including ZeroDP, achieve privacy through two forward passes with added noise, our DP-ULR distinguishes itself in several key aspects:
>
> - To obtain differential privacy, these algorithms add additional noise to the zeroth-order gradients or losses. In contrast, we explore the potential for utilizing inherent randomness "free lunch" provided by the noise injection operation in the forward learning algorithms to achieve differential privacy.
>
> - Existing methods, such as Simultaneous Perturbation Stochastic Approximation (SPSA), add noise to parameters, which often have dimensions significantly larger than logits. This leads to higher computational costs and estimation variance, especially when training from scratch. Partly because of this, those existing methods are designed for fine-tuning pre-trained models. In contrast, DP-ULR operates directly on logits, enabling the training of deep learning models from scratch and reducing the computational overhead.
>
> - ZeroDP's privacy cost scales quadratically with the number of repetitions $P$ (Theorem 4.1 in [1]), making it less scalable for larger $P$. In contrast, DP-ULR's privacy cost is independent of $P$, ensuring robust privacy guarantees and better scalability.
>
> These distinctions highlight DP-ULR's efficiency advantages, in terms of both privacy cost and computational cost, over ZeroDP and similar existing methods. We have included a detailed discussion in Sections 2.3 and Appendix A.5 of the revised manuscript.
>
> [1] Z Liu, J Lou, W Bao, Y Hu, B Li, Z Qin, K Ren. “Differentially Private Zeroth-Order Methods for Scalable Large Language Model Finetuning.” *arXiv preprint, 2024.*
>
> [2] Liang Zhang, Bingcong Li, Kiran Koshy Thekumparampil, Sewoong Oh, Niao He. “DPZero: Private Fine-Tuning of Language Models without Backpropagation.” *arXiv preprint, 2023.*
>
> [3] Xinyu Tang, Ashwinee Panda, Milad Nasr, Saeed Mahloujifar, Prateek Mittal. “Private Fine-tuning of Large Language Models with Zeroth-order Optimization.” *arXiv preprint, 2024.*

---

> ### Author Response · Authors · 2024-12-02
>
> Dear Reviewer RAbX,
>
> Thank you for your time and thoughtful review. With one day remaining in the discussion phase, we kindly request your feedback on our responses and look forward to your updated evaluation. If you have any additional questions or concerns, please feel free to let us know.

---

### Official Review · Reviewer_ZJUY · 2024-11-04

**Soundness:** 3
**Presentation:** 3
**Contribution:** 3
**Rating:** 6
**Confidence:** 3

**Summary:**

The paper proposed DP-URL algorithm, which is an improvement of DP-SGD by dynamically adjusting the injected noise into the model.

**Strengths:**

The proposed algorithm improved the accuracy of DP-SGD without sacrificing the privacy, especially in the large batch-size applications.

**Weaknesses:**

The runtime and complexity of the proposed algorithm are not well discussed. Even though DP-SGD does not require gradient clipping, it requires the computation of the Jacobian matrices, whose complexity seems to be cubic if I understand correctly.

**Questions:**

What is the complexity of one step in DP-URL?

What is the runtime of DP-URL in comparison with DP-SGD?

---

> ### Author Response · Authors · 2024-11-28
>
> **[W1 & Q2] The runtime of DP-ULR in comparison with DP-SGD**
>
> **[RW1 & RQ2]:** Thank you for your careful feedback. The runtime of our DP-ULR and DP-SGD is 23 seconds/epoch and 18 seconds/epoch on an A6000 GPU for the MNIST dataset, respectively. We kindly remind the reviewer that these reports have been included in the paper (lines 530–531).
>
> **[W1] Complexity of DP-ULR with computing Jacobian matrices**
>
> **[RW1]** Regarding time complexity, although our mathematical equations involve Jacobian matrices, we do not need to explicitly compute these matrices to calculate gradients. Instead, as detailed in the supplementary materials, the gradient computation function for each type of layer (e.g., Linear, Convolution) can be defined directly. For example, in linear layers, gradient computation is implemented using linear transformation. In convolution layers, it is implemented using convolution operations. Consequently, similar to backpropagation algorithms, the complexity of gradient computation in our DP-ULR for each layer matches the order of the forward operation.
>
> For example, in an MLP with $L$ layers and $N$ neurons per layer, the complexity of one forward pass is $O(LN^2)$. For backpropagation in SGD/DP-SGD, the backward pass complexity is $O(LN^2 + LN^2 + LN) = O(2LN^2 + LN) = O(LN^2)$. For DP-ULR, the gradient computation complexity is $O(LN + LN^2 + LN) = O(LN^2 + 2LN) = O(LN^2)$. A key distinction is that SGD/DP-SGD incurs an additional $LN^2$ term, while DP-ULR incurs an additional $LN$. This arises because backpropagation requires matrix multiplication for computing gradients of the input to each layer, while in DP-ULR, gradients are computed by multiplying the loss and the noise (as in the last two terms of Equation 4), which has a complexity of $O(N)$.
>
> Additionally, we kindly remind the reviewer that our DP-ULR includes sample-level gradient clipping to constrain sensitivity. We clarify the difference between DP-ULR and DP-SGD on this point as follows. Due to the need to clip each individual gradient, resulting in an independent backward pass for each example, DP-SGD is slower than traditional SGD. In contrast, the calculation of a single estimated gradient in a DP-ULR is inherently independent, and a single clipping can be performed without increasing the computational cost compared to ULR.
>
>
> **[Q1] Complexity of one step in DP-ULR**
>
> **[RQ1]:** For the complexity of one step in DP-ULR, in addition to the previous response, we consider an MLP with $L$ layers and $N$ neurons. The complexity depends on the parallelism setting:
>
> - **No Interlayer Parallelism:** The forward stage involves $L$ forward passes (independently perturbing $L$ layers), resulting in a total complexity of $O(L^2N^2)$. The "backward" stage (gradient computation) has a total complexity of $O(LN^2)$.
> - **Full Interlayer Parallelism:** A favorable property inherited from ULR is that the $L$ forward passes and gradient computations for $L$ layers can be executed in parallel. This reduces the forward stage complexity to $O(LN^2)$ and the gradient computation complexity to $O(N^2)$.

---

> ### Author Response · Authors · 2024-12-02
>
> Dear Reviewer ZJUY,
>
> Thank you for your time and thoughtful review. With one day remaining in the discussion phase, we kindly request your feedback on our responses and look forward to your updated evaluation. If you have any additional questions or concerns, please feel free to let us know.

---

### Official Review · Reviewer_JZt3 · 2024-11-04

**Soundness:** 3
**Presentation:** 3
**Contribution:** 3
**Rating:** 6
**Confidence:** 3

**Summary:**

The paper introduces the Differential Private Unified Likelihood Ratio (DP-ULR) method,to achieve differential privacy in deep learning through forward learning. Unlike traditional backpropagation-based methods, DP-ULR leverages the inherent randomness of forward learning to estimate gradients.

**Strengths:**

1. This work studies an important problem with some new insights.

2. Extensive theoretical analysis is provided.

3. The paper is well written.

**Weaknesses:**

1. The assumption of this work may not always hold in practice. First, is the number of repeat time K (which is similar to the number of training batches in DP-SGD) related to privacy cost? If so, we cannot assume it to be too large. In other words, the assumption stated in lines 237-238 is not hold. Second, the privacy cost also relies on the assumption of a full-rank covariance matrix of gradient proxies. If these assumptions do not hold, the privacy guarantees may be compromised.

2. DP-ULR is highly sensitive to the choice of parameters such as the noise standard deviation, batch size, and the rejection threshold. However, it is unclear how to select these parameters appropriately. This sensitivity can make it challenging to find the optimal settings for different datasets and models, potentially requiring extensive hyperparameter tuning.

3. As mentioned in the first paragraph of page 2, the original ULR mothed (Jiang et al., 2023) incorporates noise addition into ULR algorithm. Therefore, it appears that the contribution of this work is primarily focuses on the analysis of DP properties. The authors should clearly outline their contributes, highlighting the differences from the original ULR algorithm.

**Questions:**

1. In line 198, should be description be “Add noise to the l-th module’s input” (i.e., “output”-> “input”) ?

2. Please also refer to Weaknesses section for more questions.

---

> ### Author Response · Authors · 2024-11-28
>
> **[W1] Relation between $K$ and privacy cost; Potential unsatisfied assumption**
>
> **[RW1]:** We understand your concern about the relation between repeat time $K$ and privacy cost. Based on our theoretical results, when the noise scale $\sigma$ satisfies Equation (8), the privacy cost is independent of repeat time $K$. Therefore, the privacy budget does not restrict the choice of $K$.
>
> We appreciate your observation that violating the full-rank assumption of the covariance matrix could impair privacy guarantees. To address this, we proposed two remedies outlined in lines 297–301 and Appendix A.4, including adding extra noise and altering the location where noise is added. These two remedy approaches ensure that privacy guarantees remain unimpaired even if the assumption does not hold.
>
>
> **[W2] Parameter selections**
>
> **[RW2]:** Thank you for your thoughtful feedback on parameter sensitivity.
>
> Selecting parameters in DP deep learning is inherently challenging due to the privacy-utility tradeoff. For DP-ULR, we provide the following insights.
>
> - For **noise scale**,  we first kindly note that the target std level $\sigma_0$ in DP-ULR is equivalent to the noise scale in DP-SGD. For DP-ULR, the injected noise standard deviation $\sigma$ is determined by privacy requirements, specifically $\sigma_0$, indicating no need to select it.
> - As for **batch size**, our experiments demonstrate that DP-ULR performs better with larger batch sizes, implying a higher sample rate $q$. The results also exhibit greater robustness than DP-SGD with smaller batch sizes in terms of higher **target std levels**.
> - The **rejection threshold** primarily affects privacy cost without significantly impacting model performance. For datasets with $N > 10^5$, this choice is trivial as long as it is smaller than $qN$.
> - Additionally, increasing target std level $\sigma_0$ allows for reducing the **repeat time** $K$, increasing the inherent randomness leveragable for privacy.
>
>
> **[W3] Differences between DP-ULR and ULR.**
>
> **[RW3]:** Thank you for your suggestion to clarify our contributions. We appreciate your careful attention to detail.
>
> While the original ULR method involves noise injection into logits and gradient estimation, our DP-ULR introduces several key differences to ensure differential privacy. **First**, we employ a novel batch formation strategy involving sampling with rejection. **Second**, DP-ULR incorporates a privacy-controlling mechanism, including an additional forward pass without noise to compute covariance matrices and required noise scales. **Finally**, DP-ULR introduces per-example gradient clipping, which is not present in ULR.
>
> These enhancements distinguish DP-ULR as a privacy-guaranteed extension of ULR. In the initial submission, we included the explanations scattered in section 3.2. To better clarify our contribution, we have revised our introduction section to highlight the distinctions (and maintain detailed explanations in Section 3.2). Thank you again for your constructive feedback.
>
>
> **[Q1] Description revision suggestion**
>
> **[RQ1]:** Thank you for pointing out this point of clarification. In the paper, we denote $v^l$ as the $l$-th layer's output and $x^{l+1}$ as the $l+1$-th layer's input. In DP-ULR, noise is added to the $l$-th module’s output, and the perturbed output is used as the input for the next layer ($x^{l+1} = v^l + z$). This approach enables us to estimate the $l$-th layer's gradient. In Appendix A.1, we provide a list of symbols used in our paper to help readers better understand our notations and avoid confusion. We appreciate your attention to detail.

---

> ### Author Response · Authors · 2024-12-02
>
> Dear Reviewer JZt3,
>
> Thank you for your time and thoughtful review. With one day remaining in the discussion phase, we kindly request your feedback on our responses and look forward to your updated evaluation. If you have any additional questions or concerns, please feel free to let us know.

---

### Author Response · Authors · 2024-11-28
**General Response**

We thank all of the reviewers for their insightful and helpful feedback. We are pleased to note that they found:
- The problem addressed in the paper was considered novel, with a unique perspective on differential privacy in deep learning. [JZt3, RAbX, pQjz, dZ7t]
- The proposed method is well-defined and exhibits strong theoretical guarantees. [JZt3, RAbX, pQjz, dZ7t]
- Multiple reviewers appreciated the depth and thoroughness of the experiments conducted, indicating the robust methodology used in the study. [RAbX, pQjz]
- The paper is well-written, with clear explanations and organization. [JZt3, RAbX, pQjz]

We highlight key revisions and clarification following (and answer individual questions in reviewer-specific responses). **Updates are blue in our revised PDF.** We hope the reviewers consider our revisions, clarification, and individual responses and adjust their scores accordingly.

- We revised the abstract and introduction section to clarify our contributions and our differences from the original ULR.
- We add discussions in Section 2.3 and Appendix A.5 to compare our DP-ULR with existing DP zeroth-order methods.
- We repeat our experiments five times independently and report the average and standard deviation in Table. 1.
- We conduct additional experiments in Appendix C.1 to analyze the relationship between noise-redundancy impairment and model size.

---

### Meta-Review · Area_Chair_8yvC · 2024-12-22

**Metareview:**

The paper introduces a new algorithm, the Differential Private Unified Likelihood Ratio method (DP-ULR), which enhances privacy protection in deep learning by utilizing forward-learning techniques instead of traditional backpropagation. DP-ULR integrates noise during the forward pass. Theoretical analysis and experiments claim to show that DP-ULR achieves better performance than DP-SGD.

The reviewers addressed concerns such as impractical assumptions, misleading comparison to DP-SGD, and impracticality of the suggested method. This paper is on the borderline, and addressing these issues more thoroughly, I have no doubt this paper will make a great submission for the next ML conference.

**Additional Comments On Reviewer Discussion:**

The reviewers addressed concerns such as impractical assumptions, misleading comparison to DP-SGD, and impracticality of the suggested method. The authors attempted to address these issues, while some of them were not resolved, e.g., the additional experiments show the suggested method perform poorly with larger batch sizes in larger models. An additional question I have is that (the same as the reviewer RAbX) in the usual DP-SGD, the privatization of gradients make both input and output pairs privatized as a result, while the suggested method seems to privatize the inputs, only. It is unclear how the comparison to DP-SGD is valid.

---

### Decision · Program_Chairs · 2025-01-22

Reject